EMBO
Molecular Medicine

# Fine-tuning FAM161A gene augmentation therapy to restore retinal function

Yvan Arsenijevic [1,8 ✉], Ning Chang [1,2,8], Olivier Mercey[3], Younes El Fersioui[1,2],
Hanna Koskiniemi-Kuendig[1], Caroline Joubert[1], Alexis-Pierre Bemelmans [4], Carlo Rivolta [5,6],
Eyal Banin[7], Dror Sharon[7], Paul Guichard [3], Virginie Hamel [3] & Corinne Kostic [2 ✉]

## Abstract

**For 15 years, gene therapy has been viewed as a beacon of hope for inherited retinal diseases. Many preclinical investigations have centered around vectors with maximal gene expression capabilities, yet despite efficient gene transfer, minimal physiological improvements have been observed in various ciliopathies. Retinitis pigmentosa-type 28 (RP28) is the consequence of bi-allelic null mutations in the FAM161A, an essential protein for the structure of the photoreceptor connecting cilium (CC). In its absence, cilia become disorganized, leading to outer segment collapses and vision impairment. Within the human retina, *FAM161A* has two isoforms: the long one with exon 4, and the short one without it. To restore CC in *Fam161a*-deficient mice shortly after the onset of cilium disorganization, we compared AAV vectors with varying promoter activities, doses, and human isoforms. While all vectors improved cell survival, only the combination of both isoforms using the weak FCBR1-F0.4 promoter enabled precise FAM161A expression in the CC and enhanced retinal function. Our investigation into *FAM161A* gene replacement for RP28 emphasizes the importance of precise therapeutic gene regulation, appropriate vector dosing, and delivery of both isoforms. This precision is pivotal for secure gene therapy involving structural proteins like FAM161A.**

**Keywords** Ciliopathy; Gene Therapy; Promoter Activity; Retinal Degeneration
**Subject Category** Genetics, Gene Therapy & Genetic Disease

## Introduction

Inherited retinal dystrophies (IRDs) encompass a group of hereditary diseases caused by a large number of genes and mutations, leading to loss of vision for which there is currently no cure for the great majority of cases (Schneider et al, 2022). More than 300 causing genes were identified (https://web.sph.uth.edu/RetNet/). Over the last 20 years, proof-of-concept for gene therapy to restore visual function and delay retinal degeneration in different models of IRDs has been proposed (Garafalo et al, 2020; Trapani and Auricchio, 2019). From these studies, several of the vectors are being tested in clinical trials (Garafalo et al, 2020; Trapani and Auricchio, 2019). It is worth mentioning that an FDA-approved drug, Luxturna (Russell et al, 2017), paved the way for the gene therapy development for ocular diseases. Nonetheless, the application of this technology to treat certain forms of IRD either failed or produced imperfect retinal rescue. This concerns mainly structural proteins, proteins linked to cell structures or proteins necessitating an appropriate stoichiometric level with other partners to be functional, this feature being often not known. For example, gene therapy for Bardet-Biedl Syndrom-1 showed efficacy in expressing the transgene, but the BBS1 protein accumulated in the photo-receptors, and no restoration of visual function was observed (Seo et al, 2013). Deficiency of the ciliary protein LCA5 provokes retinal degeneration and gene therapy to restore its expression was only partially successful (Faber et al, 2023). Indeed, only 4 animals out of 16 treated showed improved retinal activity monitored by ERG and several others revealed improved photoreceptor responsiveness to light using multiple electrode arrays, suggesting that only patches of transduced photoreceptors expressing the right amount of the LCA5 protein properly restored photoreceptor function. Investigation using the Ultrastructure Expansion Microscopy (U-ExM) method to probe for the cellular outcome of the LCA5 gene augmentation strategy revealed that many photoreceptors showed an extension of the LCA5 protein outside of the normal localization (Faber et al, 2023), suggesting that protein mis-localization may account for impaired functional rescue. However, the success of long-term effects with other genes coding for ciliary proteins with treatment at any age of the disease was documented for RPGR or NPHP5 (Aguirre et al, 2021; Beltran et al, 2015) revealing different regulations of protein addressing to the cilium.

Retinitis pigmentosa-type 28 (RP28) is the consequence of bi-allelic null mutations in the *FAM161A* gene (Bandah-Rozenfeld

[1]Unit of Retinal Degeneration and Regeneration, Department of Ophthalmology, University of Lausanne, Jules-Gonin Eye Hospital, Fondation Asile des Aveugles, Lausanne, Switzerland. [2]Group for Retinal Disorder Research, Department of Ophthalmology, University of Lausanne, Jules-Gonin Eye Hospital, Fondation Asile des Aveugles, Lausanne, Switzerland. [3]University of Geneva, Department of Molecular and Cellular Biology, Sciences III, Geneva, Switzerland. [4]Université Paris-Saclay, CEA, CNRS, Laboratoire des Maladies Neurodégénératives: mécanismes, thérapies, imagerie, Fontenay-aux-Roses, France. [5]Institute of Molecular and Clinical Ophthalmology Basel (IOB), Basel, Switzerland. [6]Department of Ophthalmology, University of Basel, Switzerland. [7]Department of Ophthalmology, Hadassah-Hebrew University Medical Center, Faculty of Medicine, The Hebrew of Jerusalem, Jerusalem, Israel. [8]These authors contributed equally: Yvan Arsenijevic, Ning Chang. ✉E-mail: yvan.arsenijevic@fa2.ch; corinne.kostic@fa2.ch

et al, 2010; Langmann et al, 2010), which codes for a protein (FAM161A) known to form part of the inner scaffold structure maintaining microtubule doublets of the connecting cilium and the basal body/centriole (Le Guennec et al, 2020). Interestingly, FAM161A, when overexpressed in human cells, decorates cytoplasmic microtubules, highlighting its microtubule-binding ability (Di Gioia et al, 2012; Le Guennec et al, 2020; Zach et al, 2012). Interestingly among the potential homolog proteins, FAM161A has only one strong hit with more than 95% with a predicted shared structure of TPX2 (*Xenopus* kinesin-like protein-2) (Levine, 2020). It is thus not surprising that FAM161A is a microtubule-binding protein and may interact with other centriolar proteins as already described (Di Gioia et al, 2015; Le Guennec et al, 2020).

We recently showed in the *Fam161a*^*tm1b/tm1b*^ mouse deficient for *Fam161a* gene that the connecting cilium (CC) did not form properly and displayed ultrastructural aberrations with spread microtubule doublets due to the loss of the inner scaffold (Mercey et al, 2022). Moreover, we found that the localization of other proteins composing the CC such as the centriolar protein POC5 or the connecting cilium protein CEP290 was markedly reduced. In addition, the cilium was fully disorganized leading to an abnormal distribution of the Rhodopsin protein necessary for the phototransduction function (Karlstetter et al, 2014; Mercey et al, 2022). All proteins involved in the phototransduction pathway traffic through this CC structure to form or reach the outer segment to ensure light detection, emphasizing the importance of cilium integrity for photoreceptor function. Alteration of the CC in *Fam161a*^*tm1b/tm1b*^ mice leads to slow and progressive cell death, phenocopying the human RP28 disease. It is also important to note that CEP290 deficiency leads to retinal degeneration. A correct reconstruction of the CC in the *Fam161a*-associated deficiency appears thus necessary to restore cell function and maintain cell survival in the FAM161A-deficient retina. Seizing the opportunity of this mouse model which recapitulated human disease features with a relatively slow degenerative process (Beryozkin et al, 2021), we investigated the possibility of restoring the thin structure of the CC and tested different levels of gene expression to assess whether this CC formation necessitates critical control of FAM161A expression. In a previous study, the gene transfer of the long mouse *Fam161a* isoform under the control of the *Interphotoreceptor Retinoid-Binding Protein* enhancer combined with the *G Protein-Coupled Receptor Kinase 1* promoter (IRBP-GRK1), showed efficient transgene expression in rod and cone photoreceptors leading to retina rescue and improved retinal function (Matsevich et al, 2023). Nonetheless, some ectopic FAM161A expression extending outside the CC was observed. Because it is known that FAM161A overexpression leads to ectopic FAM161A localization on cytoplasmic microtubules (Di Gioia et al, 2012; Le Guennec et al, 2020; Zach et al, 2012), which could provoke potential cellular side effects, we aimed to optimize gene transfer and to translate this gene therapy approach using the human *FAM161A* cDNA.

In the human retina, *FAM161A* generates two major isoforms due to the alternative skipping of the fourth exon (Bandah-Rozenfeld et al, 2010; Langmann et al, 2010). We tested these two isoforms, individually or in combination, using AAV2/8 vectors bearing the shorter or the longer human isoform controlled by different promoters and regulatory elements. We identified the conditions (vector design, treatment plan, and dose) which allow

the rescue of retinal survival and function. The gene therapy allows not only to repair the disorganized CC, but also to restore the appropriate expression (no ectopic expression) and localization of other proteins composing the CC. With this study, we propose that for certain structural proteins the use of a weak promoter prevents side effects, and combined with an adapted vector dose ensures an optimized distribution of the vector in the targeted tissue.

# Results

## Vector designs for photoreceptor-specific expression of human long or human short FAM161A isoforms

To optimize gene transfer of the two human isoforms of the *FAM161A* gene (Appendix Fig. S1A; Bandah-Rozenfeld et al, 2010; Langmann et al, 2010), we investigated different promoters and administration possibilities. To validate the expression of these two isoforms from the cDNA sequences, each cDNA was inserted in a plasmid bearing the ubiquitous, intronless, *Elongation factor-1-alpha* promoter (EFS) (Kostic et al, 2003). FAM161A protein expression was first tested in ARPE-19 cells where the hFAM161A was detected by immunofluorescence (Fig. 1A). As previously described, we confirmed in hTERT-RPE1 cells that overexpressed long or short hFAM161A isoforms decorated the cytoplasmic microtubules and stabilized microtubules as indicated by the strong acetylated tubulin signal (Di Gioia et al, 2012; Le Guennec et al, 2020; Zach et al, 2012) (Fig. 1C). In comparison, cells that do not overexpress FAM161A presented only a few acetylated-tubulin filaments (Fig. 1C). The size of the two hFAM161A isoform products was confirmed by western blots (Fig. 1B).

After this validation, different vector constructs were generated to test gene transfer efficacy according to the promoter activity. In a previous study, the *Interphotoreceptor Retinoid-Binding Protein* enhancer combined with the *G Protein-Coupled Receptor Kinase 1* promoter (IRBP-GRK1) drove efficient transgene expression in rod and cone photoreceptors but some ectopic FAM161A expression extending outside the CC was observed (Matsevich et al, 2023). Therefore, we reasoned that using the endogenous *hFAM161A* promoter in the vector design may better mimic the endogenous expression and would represent an asset for the gene augmentation strategy. The mouse *Fam161a* promoter was shown to contain two Crx-bound-regions located in the promoter region (*Cbr1*) and in the first intron (*Cbr2*) (Langmann et al, 2010). *Cbr1* and *Cbr2* are well conserved in mammals (Langmann et al, 2010), suggesting an active role for *hFAM161A* transcription. We thus identified the corresponding region in the human gene by direct alignment centered on the ATG start site of the mouse and human genomes. We identified putative Crx-binding sites in two human regions, in a 607 bp fragment 5' of the transcription start site defined as FCBR1 and in a 457-bp fragment in the human first intron defined as FCBR2 (Appendix Fig. S1B). We also defined a 460-bp segment as the *FAM161A* core promoter (F0.4, Appendix Fig. S1B). To design an endogenous-like promoter for *FAM161A*, we fused FCBR1 or FCBR2 to F0.4 (Fig. 1D).

For all vectors, three *FLAG* sequences were added at the 5' of the hFAM161A transgene, as well as a *WPRE4* sequence in the 3' of the transgene cassette to stabilize the RNA (Schambach et al, 2006) followed by a polyA (Fig. 1D). Gene expression driven by these

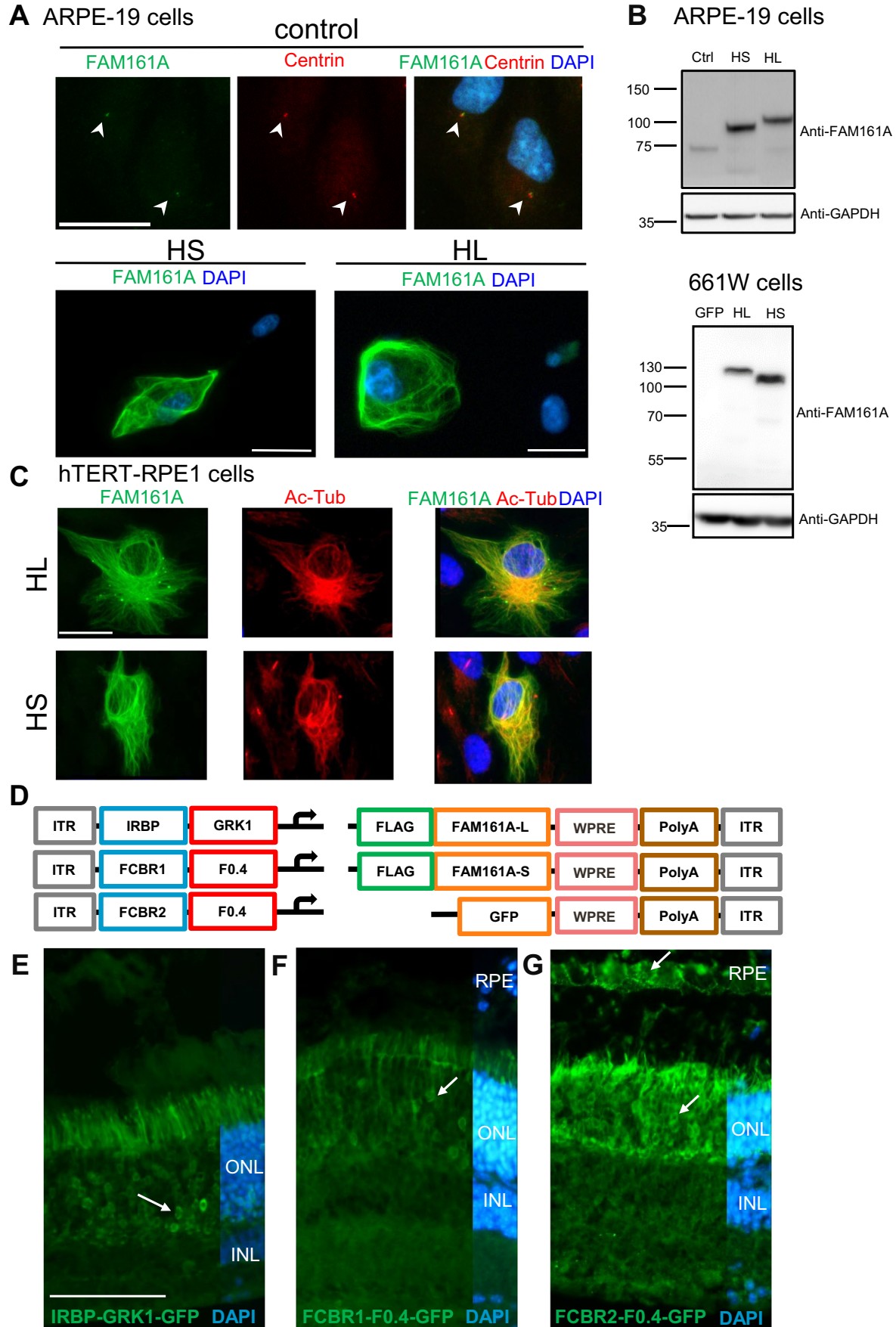

**Figure 1. AAV to generate long (L) and short (S) hFAM161A isoforms.**

(A) Untransfected ARPE-19 cells (control) were labeled for FAM161A (green) and centrin (red) with arrowheads indicating co-labeled centrioles. To validate both HS and HL FAM161A cDNAs, ARPE-19 cells were transfected with the HL or HS isoforms and labeled for FAM161A (green). The labeling of overexpressed HS and HL was strong and needed a 20-fold decrease in exposure time compared to the labeling of endogenous FAM161A in untransfected cells. (B) Western blot of ARPE-19 and 661 W cells nontransfected (ctrl) or transfected with expression plasmids for GFP, HL, or HS showed the different sizes of the two protein isoforms. (C) hTERT-RPE1 cells were transfected with either the HL or HS isoforms and labeled for FAM161A (green) and acetylated tubulin (red). Overexpression of both isoforms colocalize with acetylated tubulin. (D) Schematic representation of the different elements combined in the AAV8 vectors produced for in vivo studies. Three versions of regulatory elements were followed by either of the three types of transgene elements. (E–G) GFP expression (arrows) one-month post subretinal injection of AAV2/8-IRBP-GRK1-, FCBR1-F0.4-, and FCBR2-F0.4-GFP vectors. Scale bar in (A, C) 25 μm and (E) 100 μm. HS human short isoform, HL human long isoform, Ac-Tub acetylated Tubulin, ONL outer nuclear layer, INL inner nuclear layer, RPE retinal pigment epithelium. Data information: Results in (A) are representatives of four (ARPE-19) and two (661 W) independent experiments. One sample for each condition and each experiment was processed for immunocytochemistry. Data obtained in (C) were generated from one single experiment. Results in (E–G) are representative of 5, 12, and 5 experiments, respectively, with the corresponding eye number processed for IHC: $n = 12$, 6, and 9. Source data are available online for this figure.

different promoters was first tested with a GFP transgene without the *FLAG* tag. AAV2/8 vectors were individually administrated by subretinal injection into postnatal day (PN) 14–15 *Fam161a^tm1b/tm1b* mice, and the GFP expression was analyzed one to two months later by immunolabeling. The IRBP-GRK1 promoter at a dose of $1.3 \times 10^9$ genomic copies in 2 μL (GC/2 μL) induced a marked GFP expression in photoreceptors as expected ($n = 12$, Fig. 1E), whereas the vector with FCBR1-F0.4 promoter at a dose of $1 \times 10^{10}$ GC in 1 μL produced low levels of GFP ($n = 6$, Fig. 1F) detectable in a limited region close to the injection site in the photoreceptor layer (outer retinal layer, ONL). By contrast, the vector with the FCBR2-F0.4 promoter ($1 \times 10^{10}$ GC/0.5 μL) produced a robust expression of GFP in photoreceptors and the retinal pigment epithelial cells ($n = 9$, Fig. 1G).

## AAV2/8-IRBP-GRK1 vectors partially preserve the retina integrity, but not its function

We have previously shown that *mFam161a* expression of the longer isoform delivered by the AAV2/8-IRBP-GRK1 vector in the KO mouse retina preserves retinal structure and function (Matsevich et al, 2023). Here, we performed a similar study using the hFAM161A vectors. The AAV2/8-IRBP-GRK1-HL_FAM161A-WPRE (IRBP-GRK1-HL) or AAV2/8-IRBP-GRK1-HS_FAM161A-WPRE (IRBP-GRK1-HS) vectors ($10^{10}$ GC/μL) were subretinally injected separately into PN15 *Fam161a^tm1b/tm1b* eyes. Control eyes were injected with AAV2/8-IRBP-GRK1-GFP-WPRE (IRBP-GRK1-GFP). Surprisingly, in contradiction to the *mFam161a* vector, neither of the *hFAM161A* vectors restored or preserved retina function as attested by electroretinogram (ERG) recordings 3 months post injection (mpi) (Fig. 2A). We confirmed that FAM161A expression after 3–6 mpi was well detected in photoreceptors, and was not only localized at the CC level but also often extended throughout the outer nuclear layer and was strongly expressed in the inner segment (ONL, Fig. 2E–H). Such a pattern of expression was observed for both the HS- and the HL-transferred isoforms. At 6 mpi, the ONL thickness was around twofold thicker in the *FAM161A*-treated groups in comparison to the GFP control group or untreated region, (Fig. 2J–M). In another set of experiments, the two HS and HL vectors were co-injected at the same age (PN14-15) in the FAM161A-deficient mice ($10^{10}$ GC/μL in total, $n = 18$). No significant improvement of the ERG response under scotopic conditions was evidenced compared to the GFP control group despite a tendency observed at 2 mpi ($P = 0.0828$, $n = 18$, Fig. 2B–D). The pattern of FAM161A expression was

similar to the injection of single isoforms, and it was detected in the photoreceptors with ectopic expression in the cell body (Fig. 2I, orange arrows; Appendix Fig. S2). The retina thickness was also well preserved with this co-treatment and corresponded to more than two-thirds of the WT retina thickness Fig. 2N). To further gain insights into the distribution of FAM161A in the retinal layer, especially at the level of the CC, and to determine if abnormal features appeared with this treatment, we turned to Ultrastructure Expansion Microscopy (U-ExM) (Gambarotto et al, 2019) (Fig. 3). We found that FAM161A was present in the CC but no longer restricted to it (white arrow) and instead extended towards the inner segment as well as along the entire axoneme, stained with anti-tubulin antibodies (Fig. 3, orange arrows). The re-expression of FAM161A protein leads to the full decoration of microtubules in IS and ONL, such as seen in in vitro experiments (Di Gioia et al, 2012; Le Guennec et al, 2020; Zach et al, 2012). Nevertheless, FAM161A expression closed the opened microtubule filaments. Looking at the CC only, the length of the FAM161A, CEP290, or POC5 labeling is increased in comparison to a WT retina (Fig. 3). Treatment also restored the expression of two other proteins found in the cilium, LCA5, and IFT81, although the expression pattern was imperfect (Fig. 3, orange arrows).

We then investigated whether the extended photoreceptor expression was due to a dose-dependent effect. Two lower doses were tested with the combination of the IRBP-GRK1-HS plus IRBP-GRK1-HL vectors at $10^9$ or $10^8$ GC (total) per injection (Appendix Fig. S2). Although the density of the transduced cells markedly decreased with the lowest dose, many cells with expression of FAM161A in the photoreceptor cell body were observed. Thus, the dose reduction decreased the number of transduced cells (as expected), but did not reduce the spread localization of FAM161A in the photoreceptor cells. As expected, no functional improvement was noted in these low-dose conditions.

## AAV2/8-FCBR1-F0.4 vectors allow correct FAM161A localization, restore photoreceptor structure, and prevent retinal degeneration

We hypothesized that the lack of functional rescue with IRBP-GRK1 vectors may be due to the mis-localization of the FAM161A proteins in photoreceptor cells due to high expression of the transgene as observed in in vitro overexpression studies (Fig. 1). We thus explored the efficiency of two composite promoters, FCBR1-F0.4 and FCBR2-F0.4, derived from the human *FAM161A* gene sequences (Appendix Fig. S1) to drive

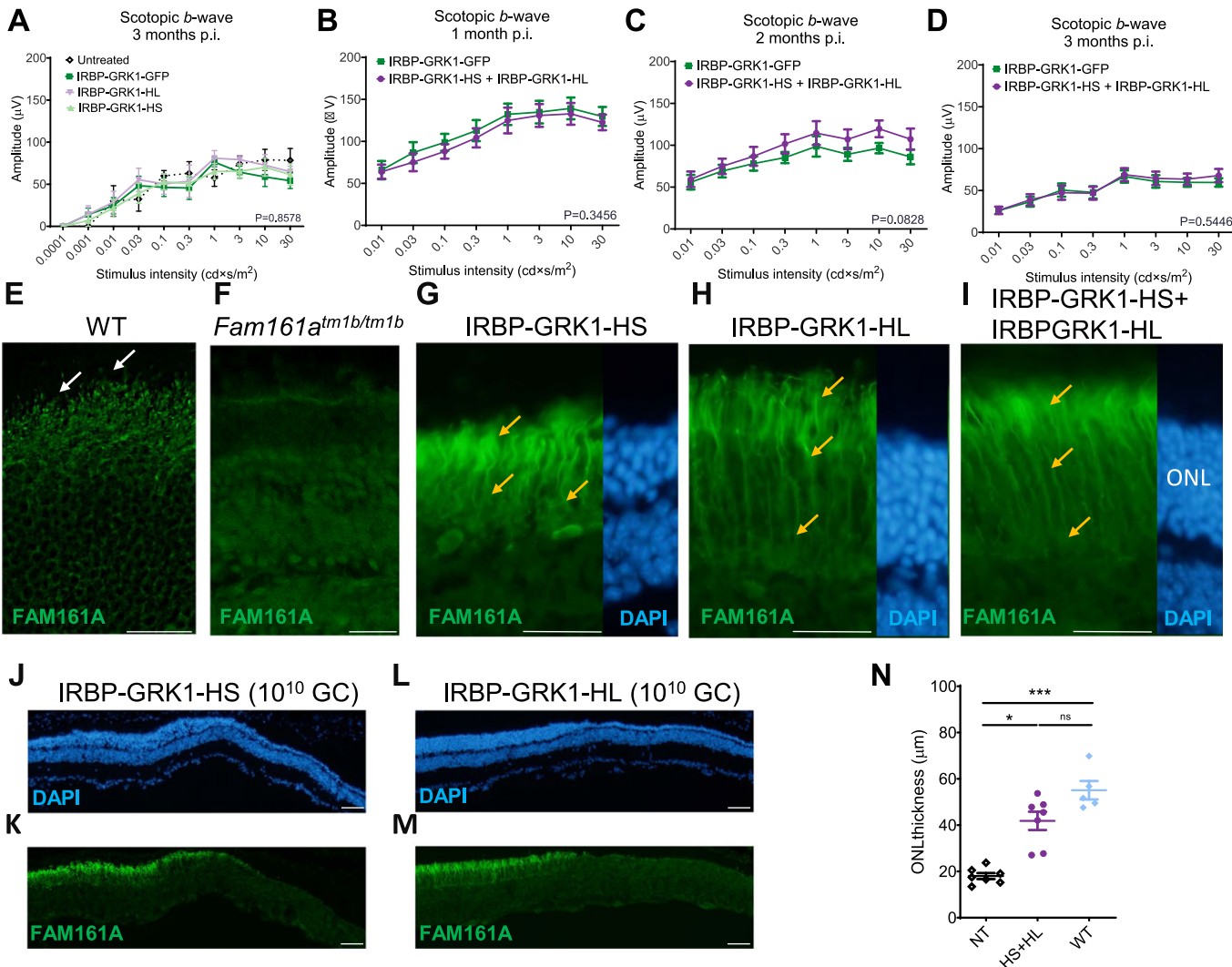

**Figure 2. AAV2/8-IRBP-GRK1-FAM161A preserved retina morphology but not function.**

(A) Retinal activity recorded by ERG in scotopic conditions 3 mpi in *Fam161a^tm1b/tm1b* mice of either IRBP-GRK1-HS or HL vectors (10^10 GC) at PN14 and in the contralateral eye with IRBP-GRK1-GFP (untreated: $n = 4$, GFP: $n = 9$, HL: $n = 6$, HS: $n = 7$, all at a dose of 10^10 GC/injection) showed no functional improvement in the hFAM161A-treated eyes. (B–D) The retinal activity was determined in *Fam161a^tm1b/tm1b* mice injected at PN14 with a mix of the two vectors (HS + HL) and monitored by ERG after 1 month (B, $n = 18$), 2 months (C, $n = 18$) and 3 months (D, $n = 12$). Data are expressed as mean + SEM. No improvement was observed at any of these time points. Expression pattern of FAM161A in WT (E), KO (F), and 6 mpi after injection with IRBP-GRK1-HS (10^10 GC, G) or –HL (H), or 3 mpi after injection with IRBP-GRK1-HS + IRBP-GRK1-HL (I) into *Fam161a^tm1b/tm1b* mouse eyes. Endogenous FAM161A expression is pointed by white arrows in (E). Note the ectopic expression of FAM161A (orange arrows, G–I). In addition, the CC is not distinguishable. (J–M) ONL thickness 6 months post injection of either HS or HL vectors. Note the thicker ONL in FAM161A-positive areas compared to FAM161A-negative region. (N) The combination of HS and HL isoforms ($n = 9$, 10^10 GC in total) enhanced retina thickness rescue in comparison to the non-treated area ($n = 9$, $P < 0.001$) to reach the WT ($n = 5$) retina thickness. Scales bars are 25 μm. ONL outer nuclear layer, HS human short isoform, HL human long isoform. A one-way ANOVA Kruskal–Wallis test was performed, followed by Dunn's test for multiple comparisons between the groups. *$P < 0.05$; ***$P < 0.001$. Data information: The result in (E) is representative of immunolabeling performed on two eyes from two individual mice. The result in (F) is representative of three eyes from three individual mice as well as of the non-treated regions of all AAV-treated eyes of the experiments analyzed in this study. Results in (G–M) are representative of five independent experiments with the respective number of eyes: HS: $n = 18$, HL: $n = 18$, and HS + HL: $n = 12$. Source data are available online for this figure.

expression in the retina. First, *Fam161a^tm1b/tm1b* mice were subretinally injected at PN14 with 10^10 GC of AAV2/8-FCBR2-F0.4-HS_FAM161A-WPRE (FCBR2-F0.4-HS) vector and analyzed one or three months later. The FCBR2-F0.4 vector induced a strong expression of the HS protein with mis-localization invading the ONL ($n = 6$, Fig. 4A). By contrast, rare discrete FAM161A labeling was observed in the CC of photoreceptors, after the injection of the AAV2/8-FCBR1-F0.4-HS_FAM161A-WPRE

(FCBR1-F0.4-HS) vector ($n = 6$, Fig. 4B), even if some additional spread expression/localization into the photoreceptor cells were also observed (Fig. 4B; Appendix Fig. S3). Interestingly, the injection of the AAV2/8-FCBR1-F0.4-HL_FAM161A-WPRE (FCBR1-F0.4-HL) vector also produced a restricted FAM161A localization in the CC with much less ectopic expression (Fig. 4C). Given the weak expression and few targeted photoreceptors induced by the FCBR1-F0.4 vectors in the CC, we

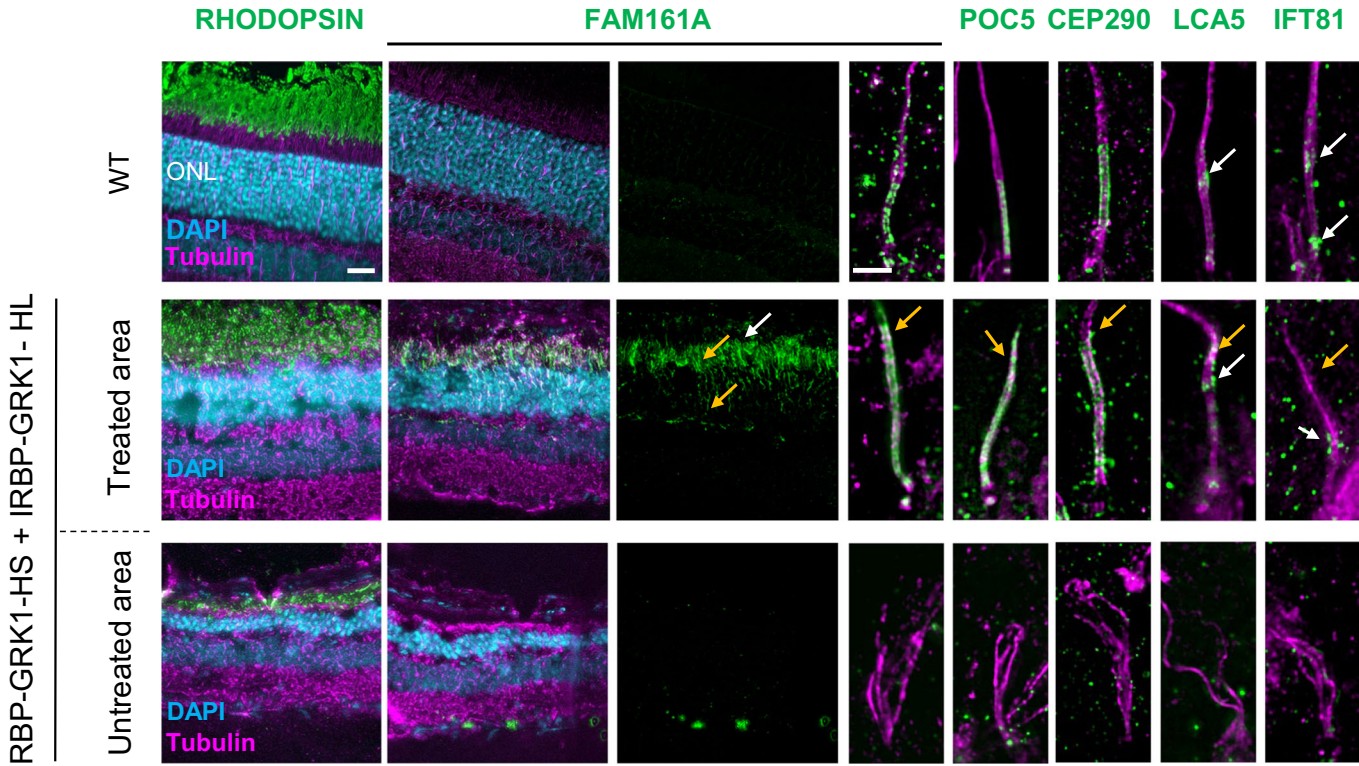

**Figure 3.   AAV2/8-IRBP-GRK1-FAM161A restores in part the CC structure.**

Ultra-tissue expansion microscopy revealed the disorganization of the KO cilium in the untreated area, whereas the co-treatment by the two isoforms restored CC structure organization in the treated area. Note that FAM161A, POC5, and CEP290 localizations are restored in the treated area but their respective labeling extends toward the apical part of the CC (orange arrows) compared to the wild type. In addition, the LCA5 labeling was not grouped, but in part dispersed (orange arrow). IFT81 was also imperfectly distributed (orange arrow) compared to the wild type. For the WT retina, FAM161A labeling from retina cryosection is hardly detectable at low magnification. Scale bar bars: 20 μm and 1 μm for low- and high magnification, respectively. ONL outer nuclear layer, CC connecting cilium. white arrows: correct labeling; orange arrows: ectopic or absence of labeling. Source data are available online for this figure.

increased the vector dose to $10^{11}$ GC/eye (Fig. 4D–G; Appendix Figs. S3 and S4). PN14 *Fam161a^{tm1b/tm1b}* mice were treated by a single injection of HL, HS, or GFP driven by FCBR1-F0.4 vectors as well as combined injection of FCBR1-F0.4-HS and FCBR1-F0.4-HL and analyzed 3 mpi. In the FCBR1-F0.4-HS ($10^{11}$ GC) condition, an increased number of FAM161A-positive CC was observed despite several photoreceptors still displaying ectopic localization of the protein in the photoreceptor cell (Fig. 4D, orange arrows). In a single injection of the FCBR1-F0.4-HL vector, an increased number of well-detectable FAM161A-positive CC was observed ($n = 7$, Fig. 4E). Interestingly, with the combined administration of FCBR1-F0.4-HL and FCBR1-F0.4-HS ($1 \times 10^{11}$ GC each), a regular and delicate FAM161A labeling was homogenously distributed in CC ($n = 7$, Fig. 4F,G; Appendix Fig. S4). Moreover, ectopic expression was observed in some horizontal-like cells (identified by their morphology and position) and in some (few) fibers of the outer plexiform layer (Fig. 4G; Appendix Fig. S3). The co-administration of HS + HL produced the best rescue of the ONL thickness in comparison to single vectors (Fig. 4H). We thus continue with this dose to investigate the potential of the vector with the FCBR1-F04 promoter.

We then investigated whether the CC structure was recon-structed after the gene therapy treatment with these different vector administrations. U-ExM was performed to investigate FAM161A precise localization and to characterize the different partners forming the CC (Fig. 5). When injecting the FCBR1-F0.4-HS

vector, we observed a closure of the CC, but FAM161A was unevenly distributed with high variations in the CC and some ectopic expression (Fig. 5B, orange arrows). A similar observation was made for its interacting protein POC5, but a shorter POC5-positive CC was measured (Fig. 5F). CEP290 and LCA5 expression was partially restored while the transporter protein IFT81 was weakly present at the level of the bulge region, where it has been shown to accumulate. The FCBR1-F0.4-HL vector also zipped the microtubule filaments at the level of the CC, and the localizations of FAM161A, POC5, CEP290, LCA5, and IFT81 resembled better the expression pattern observed in the WT CC than when treated with FCBR1-F0.4-HS (Fig. 5A,C). Importantly, for the combined injection of the two vectors coding for the HS and the HL isoforms, we found that FAM161A, POC5, and CEP290 correctly localized to the CC as well as LCA5 and ITF81 which were observed at their specific places (Fig. 5D). The non-treated area contained cilia with the absence or marked disorganization of all these ciliary proteins (Fig. 5E). The quantification of the CC length positive for FAM161A or POC5 clearly revealed that the combination of FCBR1-F0.4-HS + FCBR1-F0.4-HL restores a CC length similar to WT conditions (Fig. 5F). For comparison, CC lengths were quantified in retina treated with the IRBP-GRK1 vectors which clearly showed FAM161A and POC5 extended expressions not seen with FCBR1-F0.4 vectors (Fig. 5F). Altogether, these results

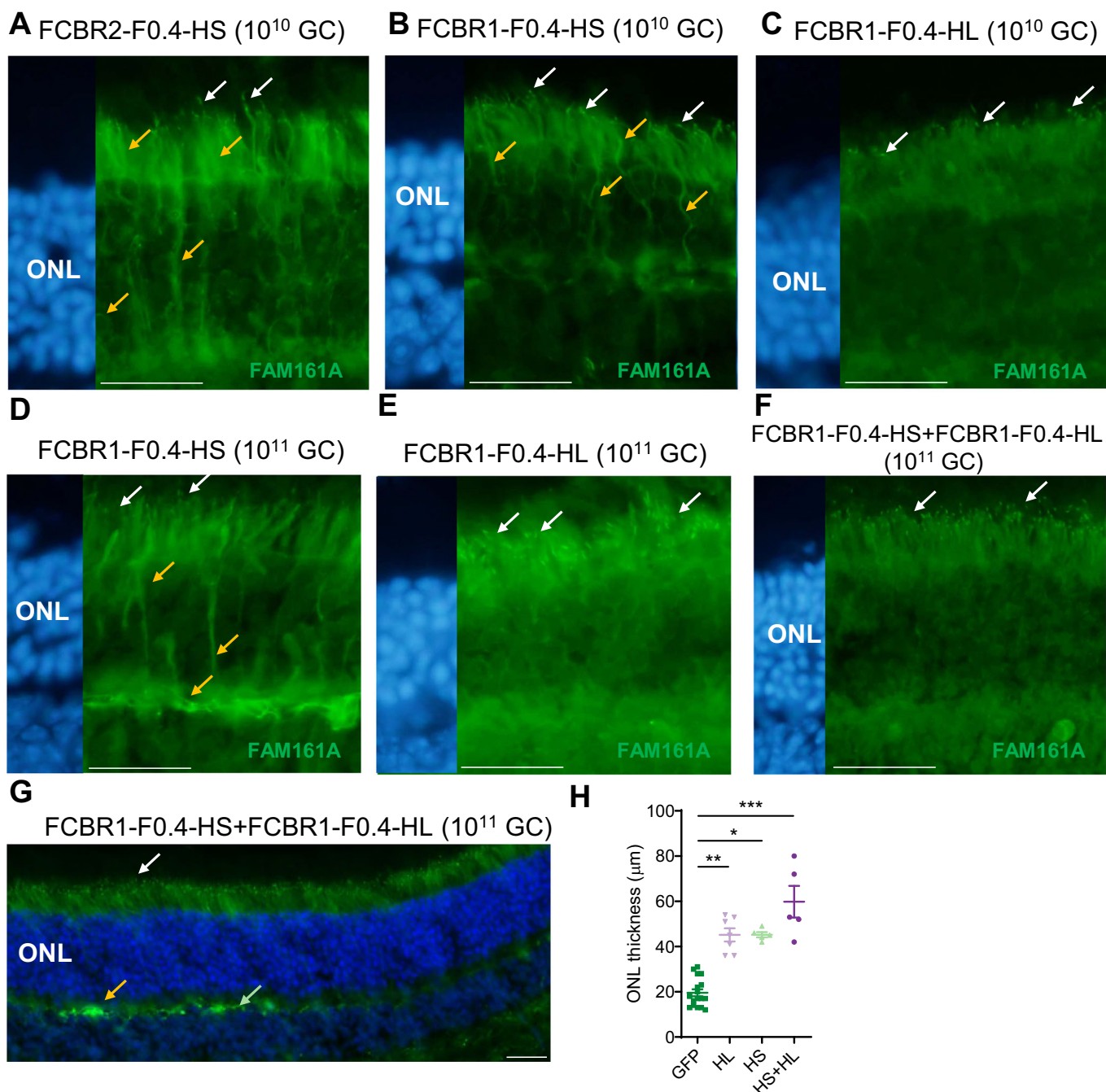

**Figure 4. AAV2/8-FCBR1-F0.4-FAM161A allowed precise FAM161A re-expression in photoreceptors cilium.**

*Fam161a[tm1b/tm1b]* mice were subretinally injected with $10^{10}$ GC of FCBR2-F0.4-HS (**A**), FCBR1-F0.4-HS (**B**), or HL (**C**) vectors between PN15 and PN26 and FAM161A expression was analyzed 2–4 months later. In total, $10^{11}$ GC were also tested for the FCBR1-F0.4-HS (**D**), and HL (**E**) vectors. Note for all conditions, the appearance of FAM161A-positive CC (white arrows), but the expression of the HS isoform induced ectopic expression of the protein (orange arrows). (**F, G**) The co-injection of FCBR1-F0.4-HS and FCBR1-F0.4-HL produced a targeted and homogenous expression of the FAM161A in CC (white arrows) with some expression in the OPL as previously described in WT (green arrow), but also sometimes stronger labeling (orange arrow). (**H**) The vector combination (HS + HL) rescued the best ONL thickness compared to GFP-treated animals ($n = 16$, $P < 0.001$). FCBR1-F0.4-HS ($n = 5$) or HL ($n = 7$) also significantly protected the ONL against degeneration (mean ± SEM, $P ≤ 0.05$, $P < 0.01$, respectively). One-way ANOVA Kruskal–Wallis test followed by Dunn's test for multiple comparisons between the groups; *$P < 0.05$; **$P < 0.01$; ***$P < 0.001$. Scale bars are 25 µm. ONL outer nuclear layer, HS short isoform, HL human long isoform. white arrows: correct labeling; orange arrows: modified labeling. Data information: Results in (**A–G**) are representative of five (FCBR2-F0.4), and five (FCBR1-F0.4-HS $10^{10}$ GC), three (FCBR1-F0.4-HL $10^{10}$), three (FCBR1-F0.4-HS $10^{11}$), four (FCBR1-F0.4-HL $10^{11}$), and three (FCBR1-F0.4-HS + HL $10^{11}$), independent experiments, $n = 5$ eyes to 16 per group as described in (**H**). Source data are available online for this figure.

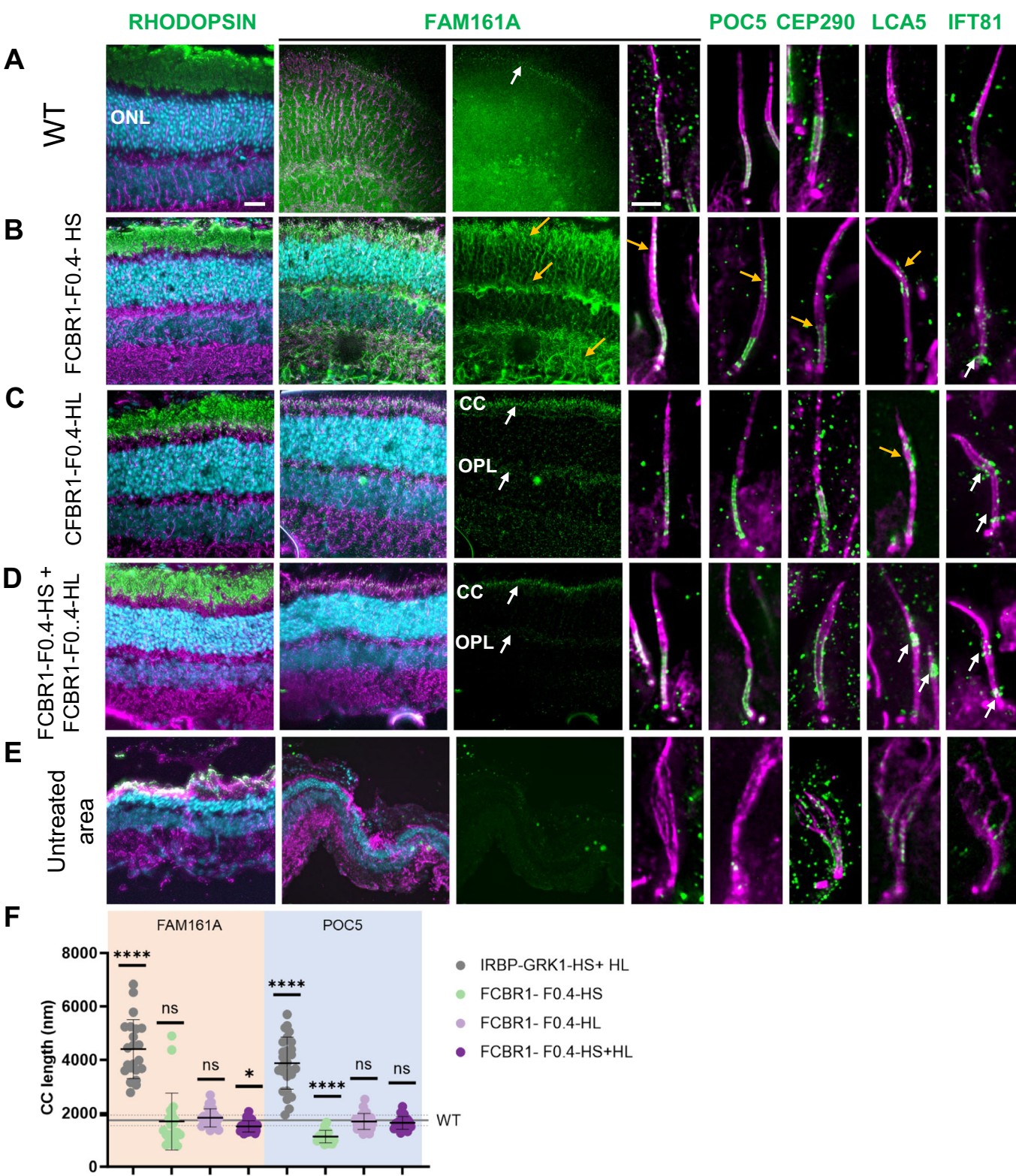

uncovered that the combination of HS + HL prevented the ectopic expression of the HS isoform and restored proper CC.

The correct subcellular addressing of the FAM161A protein in the CC resulted in remarkable protection of the ONL thickness.

Indeed, the non-transduced retina areas were markedly thinner in all groups as well as in the transduced region of the GFP control group (Figs. 5 and 6A). The quantification of the effects of the different isoforms on retina thickness revealed that the

**Figure 5.  Co-expression of long and short isoforms produced with weak promoter and high vector dose restored cilium molecular structure.**

U-ExM revealed the efficacy of the gene therapy treatment 3 months post injection. (A) Localization of proteins composing the CC in the WT retina. (B) Injection of FCBR1-F0.4-HS produced a marked rescue of the ONL (compared with (E)), but also ectopic expression of FAM161A and POC5 (see yellow arrows). Note the small size of the CEP290-positive CC (yellow arrow). The LCA5 labeling is discrete (yellow arrow), whereas the IFT81 labeling is correct (white arrows). (C) FCBR1-F0.4-HL also protected the retina and produced a correct localization of FAM161A at the CC and OPL levels as previously described (white arrows), but FAM161A labeling appeared too long in comparison to the WT CC (A, F). LCA5 is imperfectly expressed in the CC (orange arrow), but IFT81 appeared at the correct localization (white arrows). (D) The co-administration of the two FAM161A isoforms led to ONL protection, discrete FAM161A expression in the CC and OPL (white arrows), and correct localization of FAM161A, POC5, CEP290, LCA5 as well as IFT81 (white arrows). (F) Quantification of FAM161A- and POC5-positive CC lengths. Note that the CC is longer for the IRBP-GRK1-HS + IRBP-GRK1-HL group ($P < 0.0001$) and shorter for the FCBR1-F0.4-HS group ($P < 0.0001$) compared to the WT CC length determined by POC5 staining (mean and SD represented as horizontal plain and dotted line, respectively). One-way ANOVA Kruskal–Wallis test followed by Dunn's test for multiple comparisons of all groups to the control WT group; $*P < 0.05$; $****P < 0.0001$. (A) Low magnification FAM161A labelings from flatmount retina are shown, whereas all other panels are stainings on cryosection. Scale bars: 20 μm and 1 μm for low and high magnification, respectively. HS human short isoform, HL human long isoform, ONL outer nuclear layer, CC connecting cilium, OPL outer plexiform layer. White arrows: correct labeling; orange arrows: modified labeling. Data information: Results in (A–E) are representative of two sections (A, C–E) or three sections (B) from one eye for each group. Results in (F) using FAM161A labeling were obtained from measurements of 22 connecting cilia for IRBP-GRK1-HS + HL, 21 for FCBR1-F0.4-HS, 24 for FCBR1-F0.4-HL, and 27 for FCBR1-F0.4-HS + HL. Results in (F) using POC5 labeling were obtained from measurements of 27 connecting cilia for IRBP-GRK1-HS + HL, 16 for FCBR1-F0.4-HS, 28 for FCBR1-F0.4-HL, 24 for FCBR1-F0.4-HS + HL and 70 for WT. Source data are available online for this figure.

co-administration of the HS and HL isoforms was the most efficient in maintaining the ONL thickness compared to HS or HL alone (Fig. 6A). We also examined in the HS + HL group whether the rescued retina contained well-structured cone CC. Low- and high-magnification images with U-ExM revealed that cones were also re-expressing FAM161A in the rescued retina and that the connecting cilium appeared restored (Appendix Fig. S5). We then determined whether the retina rescue was dependent on the number of CC expressing a correctly addressed FAM161A protein. Figure 6B shows that, for all groups tested, the ONL thickness increased with increasing density of CC positive for FAM161A. Moreover, the HS and HL co-administration generated the highest CC density in the treated areas close to WT levels ($P > 0.9999$, $n = 4$). A clear correlation between FAM161A-positive CC density and ONL thickness can be established (Fig. 6C). A linear regression with a higher slope significantly distinguishes the HS + HL group (slope = 0.5267; $R^2 = 0.4467$) from the HS (slope = 0.3224; $R^2 = 0.5925$) or HL (slope = 0.1885; $R^2 = 0.2147$) groups ($P < 0.01$). In summary, regions with high CC density correspond to regions with the thickest ONL and the co-expression of the HL and HS isoforms potentiates the yield of FAM161A-positive CC number.

## Combined FCBR1-F0.4-HS + FCBR1-F0.4-HL administration restores retina activity in Fam161a[tm1b/tm1b] mice and prevents retinal degeneration

To reveal whether the correct protein re-expression, localization, and preservation of the retina integrity ameliorated the retina activity of *Fam161a*[tm1b/tm1b] eye mice after gene therapy, ERG recordings were monthly performed until 3 mpi. The five best-treated eyes for HS, HL and HS + HL group were selected as well as their respective GFP-injected fellow eyes. In scotopic conditions, all three treatment groups, HS, HL, and HS + HL, gave a significant increase in the *b*-wave amplitude ($P = 0.01010$, $P = 0.0009$, and $P = 0.0039$, respectively, Fig. 7A–C). Remarkably, only the combined administration of the FCBR1-F0.4 vectors encoding for HL and the HS isoforms significantly differed from the control group in dim light (0.001 cds/m²) and markedly enhanced more than twofolds the maximal response at 10 cds/m² (Fig. 7C,D). The *a*-wave response, which represents light activation of the photo-receptors showed a tendency to increase in the HS + HL treated group, although the difference did not reach a significative

threshold ($P = 0.0573$, Fig. 7E). In photopic conditions dedicated for isolating cone response, a limited effect of the HS + HL treatment was observed with a small but significant increase of the *b*-wave amplitude ($P = 0.046$, Fig. 7F) but not of the *a*-wave amplitude (Fig. 7G).

In summary, the use of a newly designed weak promoter balanced with an appropriate vector combination and dose enabled (i) the correct FAM161A protein localization in the CC, (ii) the zipping of disorganized CC of the *Fam161a*[tm1b/tm1b] retina, (iii) the restoration of CC components at their correct localization, (iv) the preservation of photoreceptor integrity preventing thus retinal degeneration, and finally (v) the improvement of retinal function. The best efficacy for human *hFAM161A* vector treatment in *Fam161a*-deficient mice was obtained only with the combined expression of the HL and the HS isoforms (Table 1).

## Discussion

Our previous study demonstrated improved retinal survival and preservation of retinal function in *Fam161a*[tm1b/tm1b] mice after gene transfer of the mouse *Fam161a* cDNA using AAV2/8-IRBP-GRK1 vector (Matsevich et al, 2023). In the present study, to translate this work into a FAM161A gene therapy for a future clinical application, we further examined the efficiency of AAV vectors in delivering the human *FAM161A* cDNA. Using conventional approaches to specifically express the transgene of interest in cones and rods with an IRBP-GRK1 promoter construct and a vector dose of about 10¹⁰ GC/injection (and lower doses as well), the human FAM161A protein was localized not only in the connecting cilium but along the cytoplasmic microtubules and also extended into the axoneme. The expression in the ONL resembles the expression pattern seen after in vitro transfection of cell lines (Fig. 1). FAM161A protein was already shown to tightly interact with microtubules and to decorate these structures when overexpressed (Di Gioia et al, 2012; Le Guennec et al, 2020; Zach et al, 2012). Thus, a controlled expression of FAM161A is required to correctly reconstruct the CC inner scaffold without ectopic cellular expression thus maintaining the cohesion of axonemal micro-tubules. For many gene therapy approaches, a strong expression is required to cover the needs of the cell or the neighboring tissue. This strategy is appropriate for proteins with high demands such as

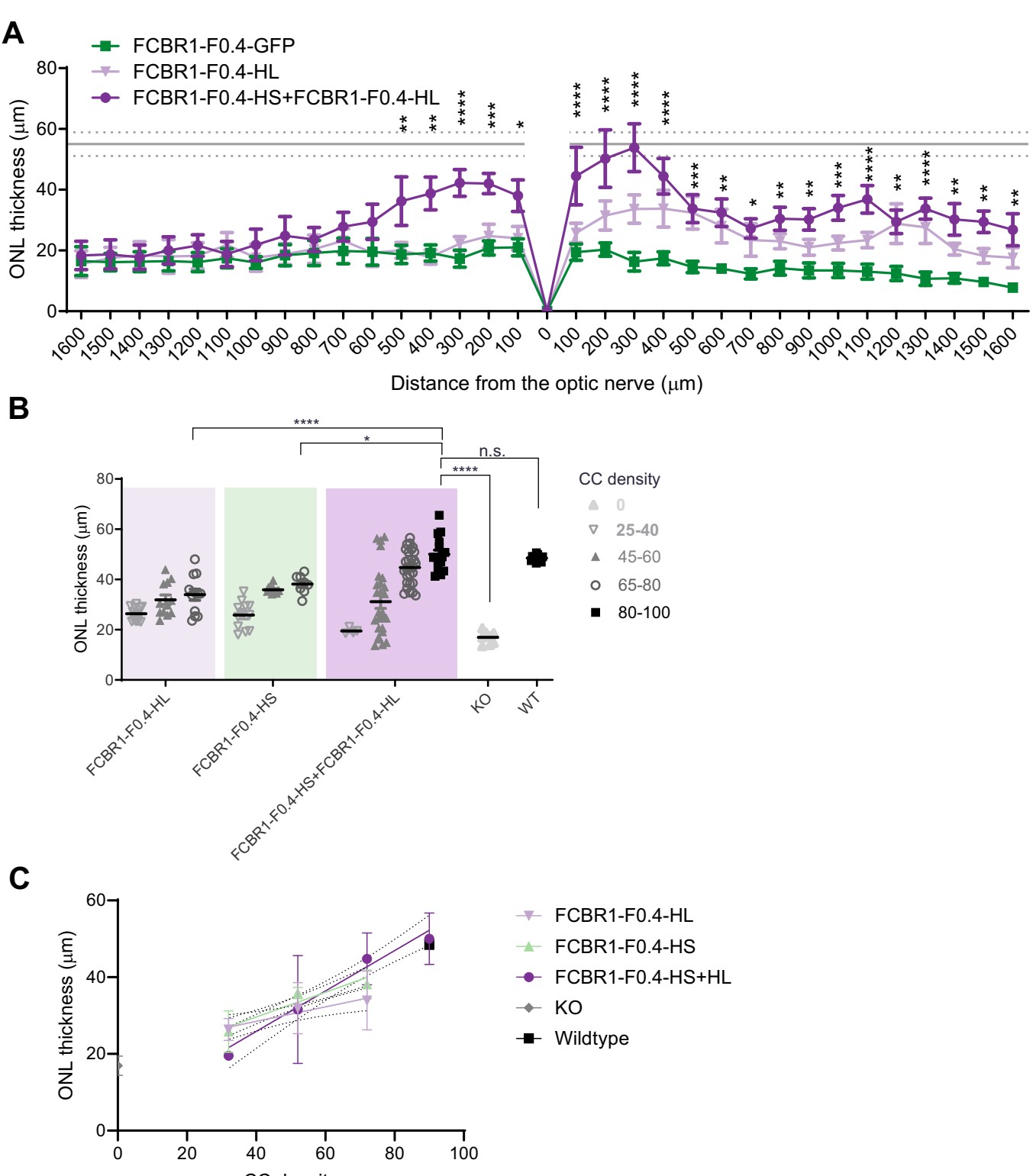

enzymes (RPE65 in RPE cells (Acland et al, 2005; Bainbridge et al, 2015) or PDE6B (Nishiguchi et al, 2015)) or necessitating renewal such as in muscles (Banks et al, 2010; Howard et al, 2021) or secreted factors (Zhang et al, 2021), but this high level of expression need is less clear for proteins constituting a structure. Our data

suggest that structural proteins such as FAM161A necessitate restricted expression to reestablish their correct function. These results suggest that permanent structures such as the cilium necessitate a low protein expression level to maintain its integrity because the renewal of the protein is not expected to occur rapidly.

◀ **Figure 6. AAV2/8-FCBR1-F0.4-FAM161A protected the retina against degeneration.**

(A) Quantification of ONL thickness throughout the whole retina revealed that the combination of HS + HL ($n = 5$) was the most efficient to rescue the retina in comparison to the GFP group ($P$ varies from <0.05 to 0.0001) or the HL group ($n = 5$). The mean (gray line) and SD (dotted gray lines) of the ONL thickness in the WT retina are indicated as reference. A two-way ANOVA followed by a Tukey's test to determine significant differences at specific locations between the FAM161A and GFP-treated eyes was performed. (B) The ONL thickness rescue by FCBR1-F0.4-FAM161A treatment was dependent on the density of FAM161A-positive CC. The highest FAM161A-positive CC density (nb of CC/200 μm) and the thicker ONL rescue were observed in the retina treated with the combination of FCBR1-F0.4-HS + FCBR1-F0.4-HL. One-way ANOVA followed by a Tukey's test for multiple comparisons of all the groups. $n = 5$ eyes per group, and 13–22 regions of 200 μm per eye were analyzed. (C) The relation between CC density and ONL thickness can be described by a linear regression (plain line). The slope of FCBR1-F0.4-HL + HS linear regression (slope = 0.5267; $R^2 = 0.4467$) is higher than the slope of FCBR1-F0.4-HL (slope = 0.1885; $R^2 = 0.2147$) or FCBR1-F0.4-HS (slope = 0.3224; $R^2 = 0.5925$) ($P < 0.01$). KO $Fam161a^{tm1b/tm1b}$, WT wild type, ONL outer nuclear layer, CC connecting cilium. Data are expressed as mean ± SEM. *$P < 0.05$, **$P < 0.01$, ***$P < 0.001$, ****$P < 0.0001$. Source data are available online for this figure.

Similarly, for proteins involved in the ciliary transport, such as BBS1 or BBS10, high expression appears to be not appropriate to restore rod physiological function (Seo et al, 2013) or efficient to restore rod-mediated vision (Hsu et al, 2023). In this case, overexpression of these proteins could saturate the transport system impacting the cellular processes. Nonetheless, other ciliary proteins involved in the transport through the cilium, such as RPGR, NPHP5, and RPGRIP1, need a higher level of expression to restore retina activity (Aguirre et al, 2021; Beltran et al, 2015; Pawlyk et al, 2005). All these results indicate that testing promoters with different activities is crucial for determining the optimal gene transfer conditions for several types of proteins.

In this work, the use of the FCBR1-F0.4 weak promoter has two main advantages. First, it allows the fine modulation of the FAM161A protein expression by adjusting the vector dose. The number of FAM161A-positive connecting cilium and their morphology improved with the vector dose. Second, it minimizes ectopic expression at the dose required to reach a density of FAM161A-positive CC similar to the WT retina. This morphological restoration in many photoreceptors results in a major rescue of the photoreceptor survival and function. On the contrary, with the IRBP-GRK1 constructs, the FAM161A protein is always ectopically expressed, even when reducing the dose by 100 times. In this case, much less photoreceptors are targeted, but the protein is also expressed in the inner segment and even in the cell body. The real consequence of this protein mis-localization is in part unclear, and even if it improves cell survival, poor functional rescue can be obtained. The mechanism of this dichotomous rescue, good survival with very limited functional effect, still needs to be unraveled.

In this study, we tested a high-dose condition to highlight the advantage of using weak promoters to treat certain forms of ciliopathy. With the optimization of vector production and purification as required for clinical applications, the final suited dose to deliver should be much less than documented here. Moreover, our strategy implies the administration of a mix of two vectors encoding each one of the two different isoforms. In this study, we just tested a one-to-one ratio, which is probably not mimicking the cell physiology. Indeed, analyses from total human retinal extract revealed that the short *FAM161A* isoform is more abundant than the long one (Langmann et al, 2010). It would be thus theoretically feasible to reduce the dose for one vector to better mimic the physiological conditions and reduce the final amount of injected viral particles. The optimal dose of each vector remains thus to be determined in future studies, as well as the scale-up in larger eyes, to bring these vectors to the clinic.

Although the short isoform is around 20-fold more abundant in the retina in comparison to the long one, the long isoform is mostly expressed in the retina in comparison to all other organs (Langmann et al, 2010). Interestingly, the injection of the vector coding only for HL produced a clear appearance of FAM161A-positive CC and weak expression in some neuronal fibers in the OPL mimicking the expression pattern of FAM161A we have previously documented (Langmann et al, 2010). The expression of the short isoform alone was also detected in the CC (but less clearly than the HL), however presented ectopic expression in the photoreceptor body and fibers connecting the OPL. In addition, the expression was strong in some horizontal cells (identified by their location and morphology). The present work focused exclusively on the gene therapy effect of FAM161A expression on cilium structure reorganization, because it is the obvious link of cilium structure defect with retinal degeneration (Mercey et al, 2022). Nonetheless, FAM161A interaction studies by yeast two-hybrid system revealed that FAM161A may also have a role in the Golgi-centrosomal interactome (Di Gioia et al, 2015). This network of proteins is involved in intracellular transport and FAM161A interacts both with proteins regulating the cytoskeletal transport and with proteins of the Golgi apparatus. Our immunolabelling results suggest that the short isoform could interact with such a protein network but less probably the long isoform, which was almost exclusively detected in the CC (see Appendix Fig. S2). The combined expression of both isoforms produced an unexpected result by presenting a major reduction of ectopic expression of the FAM161A protein in the cell body and axon accompanied by a well-structured CC. FAM161A was also reduced in neural fibers in the photoreceptors and the OPL in comparison to HS alone. These data suggest that interactions between the long and the short isoforms could affect FAM161A distribution and thus could impact FAM161A function.

Interestingly, RNAseq studies throughout the human retina development showed that alternative splicing activity occurs during retinogenesis and that isoform generations are mainly linked to RNAs coding for CC proteins, including FAM161A (Mellough et al, 2019). The combined gene transfer of the two isoforms renders thus available the protein isoforms that could be needed for the different developmental stages of the CC rebuilding and homeostasis, ensuring the photoreceptor's viability and function. Administration of appropriate isoform or isoform mix might also be required for other diseases linked to genes submitted to alternative splicing.

The cilium reorganization is probably the major action that accounts for the survival effect of the treatment and the function restoration of the combined HS + HL administration since both

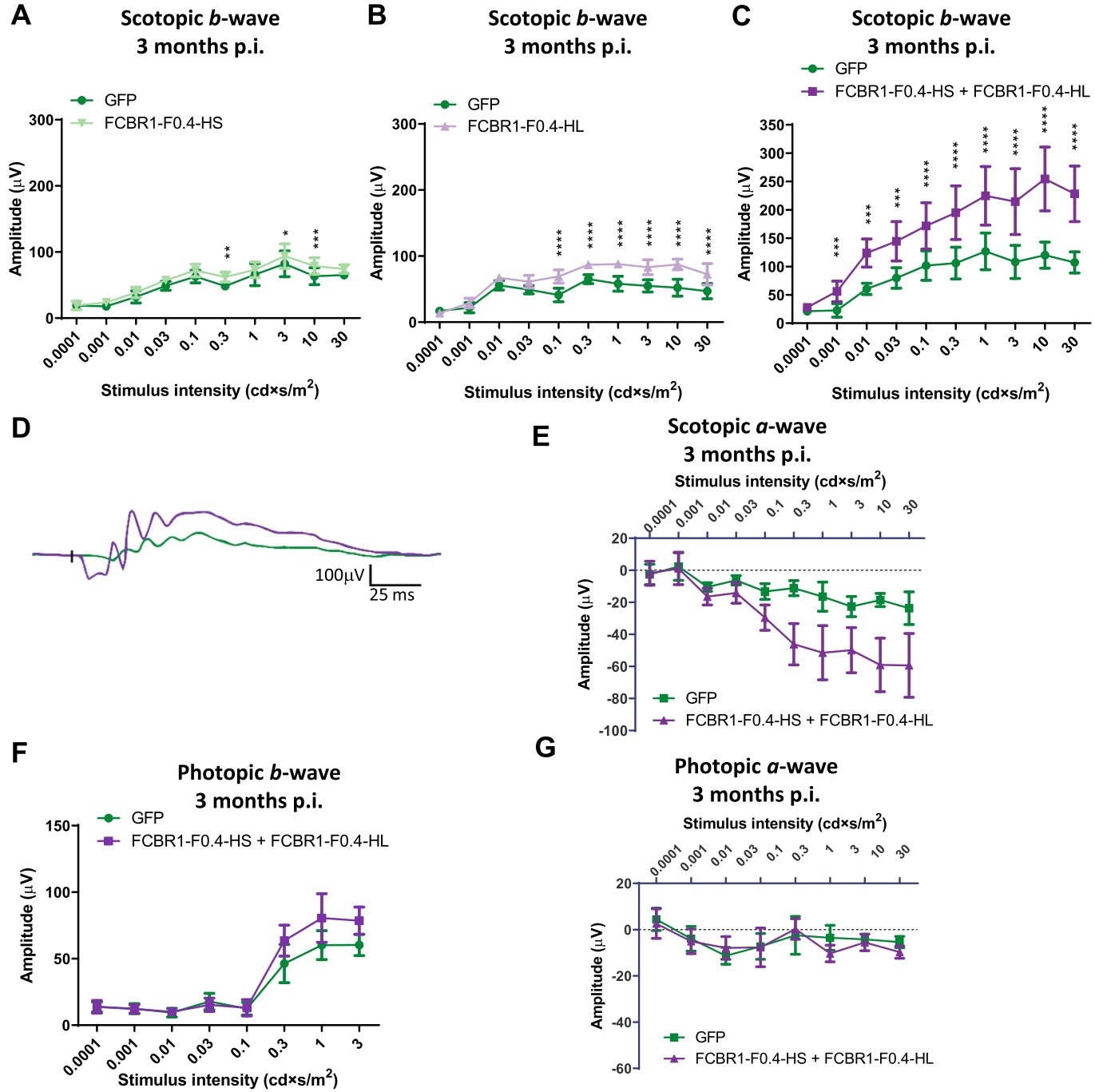

**Figure 7. Co-expression of long and short isoforms led to the best retina function rescue.**

Graph presenting retinal activity determined by ERG recordings at 3 mpi of *Fam161a*<sup>tm1b/tm1b</sup> mice subretinally injected at PN14 with $10^{11}$ GC of FCBR1-F0.4-HS (**A**), FCBR1-F0.4-HL (**B**) or FCBR1-F0.4-HS + FCBR1-F0.4-HL (**C**). FCBR1-F0.4 vectors improved the scotopic *b*-wave amplitudes to a different extent. Co-administration of both HS and HL isoforms enhances the best the *b*-wave amplitude compared to GFP-injected fellow eyes (two-way ANOVA, $P = 0.0455$), whereas HS or HL isoforms alone had a weak but significant increase in *b*-wave amplitudes (two-way ANOVA, $P = 0.0101$ and $P = 0.0009$, respectively, $n = 5$ eyes for all groups. Data are expressed as mean ± SEM). (**D**) Superposition of a representative scotopic single ERG response to 10 cd×s/m² of an *HS + HL* FAM161A-treated retina (violet) and its fellow eye treated with GFP (green). (**E**) Scotopic *a*-waves responses show a tendency to be improved by the co-treatment ($P = 0.057$). (**F, G**) The effect of co-administration of HS and HL isoform on the *b*-wave ($P = 0.046$ for the highest amplitudes) (**F**) and *a*-wave (**G**) amplitudes in photopic conditions is limited ($n = 5$ eyes for all groups. Data are expressed as mean ± SEM). Two-way ANOVA followed by Sidak's multiple comparison test for comparison of the treated eyes to the fellow GFP-treated eyes for each stimuli intensity. *$P < 0.05$, **$P < 0.01$, ***$P < 0.001$, ****$P < 0.0001$. Data information: In total for four different experiments, 12, 10, and 11 eyes were injected for the HL, HS, and HS + HL groups, respectively. The five best ERG responses per group were then analyzed. Source data are available online for this figure.

**Table 1. Summary of the effect of the different vectors after subretinal injection in *Fam161a^tm1b/tm1b* mice.**

| Vector | | | Expression | | | | Rescue | | | |
|---|---|---|---|---|---|---|---|---|---|---|
| Regulatory elements | cDNA | Dose (GC/eye) | IHC labeling | Figure | U-exM | Figure | ONL thickness | Figure | ERG | Figure |
| IRBP-GRK1 | GFP | $1.3 \times 10^9$ | Entire PR/+++ | 1E | Nd | | Nd | | Nd | |
| IRBP-GRK1 | HS | $1 \times 10^{10}$ | Entire PR/+++ | 2G | Nd | 2J | Improved | | Not improved | 2A |
| IRBP-GRK1 | HL | $1 \times 10^{10}$ | Entire PR/+++ | 2H | Nd | 2L | Improved | | Not improved | 2A |
| IRBP-GRK1 | HS + HL | $1 \times 10^{10}$ | Entire PR/+++ | 2I | Longer CC | 3, 5F, App. Fig. 4 | Improved | 2N | Not improved | 2B–D |
| IRBP-GRK1 | HS + HL | $1 \times 10^9$ | Entire PR/++ | App. Fig. 2 | Nd | | Nd | | Nd | |
| IRBP-GRK1 | HS + HL | $1 \times 10^8$ | Entire PR/+ | App. Fig. 2 | Nd | | Nd | | Nd | |
| FCBR2-F0.4 | GFP | $1 \times 10^{10}$ | PR+RPE/++ | 1G | Nd | | Nd | | Nd | |
| FCBR2-F0.4 | HS | $1 \times 10^{10}$ | Entire PR/++ | 4A | Nd | | Nd | | Nd | |
| FCBR1-F0.4 | GFP | $1 \times 10^{10}$ | PR/+ | 1F | Nd | | Not improved | | Not improved | 7A–C |
| FCBR1-F0.4 | HS | $1 \times 10^{10}$ | CC + entire PR/+ | 4B | Nd | | Nd | | Nd | |
| FCBR1-F0.4 | HS | $1 \times 10^{11}$ | CC + entire PR/+ | 4D, App. Fig. 3B | CC + entire PR | 5B,F, App. Fig. 4 | Improved | 4H, 6 | Not improved | 7A |
| FCBR1-F0.4 | HL | $1 \times 10^{10}$ | CC/+ | 4C | Nd | | Nd | | Nd | |
| FCBR1-F0.4 | HL | $1 \times 10^{11}$ | CC/++ | 4E, App. Fig. 3A | CC | 5C,F, Aupp. Fig. 4 | Improved | 4H, 6 | Weakly improved | 7B |
| FCBR1-F0.4 | HS + HL | $1 \times 10^{11}$ | CC/+++ | 4F,G, App. Fig. 3C | CC | 5D,F, App. Fig. 4 | Improved | 4H, 6 | Improved | 7C–G |

+ weak expression, ++ medium expression, +++ high level of expression, CC connecting cilium, PR photoreceptor, *nd* not determined.

isoforms are detected in the CC. Indeed, with the reappearance of FAM161A in the CC, POC5, LCA5, and CEP290 found their correct location. As important, IFT81 was relocated at its two normal sites, just above the basal body and at the bulge level (Faber et al, 2023). IFT81 is involved in the intraflagellar transport and thus contributes to the ciliogenesis (Bhogaraju et al, 2013) which may also participate to the restoration of the well-formed axonemes after gene transfer treatment as previously observed in another ciliopathy model (Faber et al, 2023). The cilium reconstruction led to a re-establishment of the RHODOPSIN transport toward the outer segment as attested by reduced immunolabelling in the ONL.

Nonetheless, gene therapy of the mouse long *Fam161a* isoform delivery ($0.5 \times 10^{11}$ GC/injection) was sufficient to maintain the retina activity and to protect photoreceptors against degeneration in the *Fam161a^tm1b/tm1b* retina (Matsevich et al, 2023). In that case, the IRBP-GRK1 promoter was used and some ectopic expression was observed in several retinal cells. However, the spread of the ectopic expression in photoreceptor cells was never comparable to the one observed using human FAM161A isoforms. Looking at this publication, several inner segments were FAM161A-positive and FAM161A labeling can be observed between the inner segment and the cell body only in a few cells (Matsevich et al, 2023). Our data would suggest that in this previous study, only a group of cells which expressed the FAM161A protein at the right level recovered their function. This would explain why little differences exist in the long term between retinas treated in one or two locations (double injections). Altogether, we propose that first the re-expression of the FAM161A protein in the photoreceptors is the major beneficial effect of the gene therapy and second, the mouse and human isoforms differently behaved in the mouse photoreceptors. The amino acid sequence homology between mouse and human is about 60% showing the diversity acquired during the evolution and potential differences in the cellular processes. Many studies remain to be undertaken to understand how FAM161A is directed either to the cilium or implicated in intracellular transport. Nonetheless, it is thus remarkable that despite the divergence between the mouse and the human isoforms, the human isoforms reconstruct the CC in the mouse *Fam161a^tm1b/tm1b* photoreceptor as described in our study.

Methodologically, our data also highlight the benefit of U-ExM observations to validate gene therapy efficacy. Faber et al (Faber et al, 2023), already used this technic to show a partial rescue of photoreceptor outer segment and axoneme after gene therapy in the *Lca5^tg/tg* mouse. However, their approach was prospective with full eyes as starting materials dedicated for U-ExM. In our study, we adapted the technic to eye sections that can be used in retrospective studies which allows us to study the same region (proximal sections) with alternative analysis. Merging the high resolution of cellular structure with the power of immunolabeling specificity on standard tissue sections, this innovative microscopy modality will certainly help in the future to better understand biological mechanisms and give an unprecedented opportunity to evaluate therapy outcomes in preclinical studies or analyze clinical samples.

In conclusion, exploring the possibility of reconstructing the CC in the FAM161A-deficient mouse retina by gene therapy, we identified that structural restoration can be achieved when a weak promoter is used to drive the expression of the transgene and combined with an appropriate dose of vector to efficiently target the tissue. This optimized approach permits to not submerge the

targeted cells with the therapeutic protein while reaching a maximal number of cells. Such a strategy is probably valid for several other applications necessitating a restricted level of the protein of interest. In addition, we revealed that the two human isoforms of the FAM161A gene are necessary for proper cilium reconstruction and function proposing a novel paradigm of treatment based on combined administration of two therapeutic sequences.

The follow-up of 100 RP28 patients for several years indicated a slow degenerative process of the retina with preserved large area around the macula until the fourth decade of age (Beryozkin et al, 2020). The vision decline is thus slow, thanks to the maintenance of the central vision. A gene therapy treatment with the vector combination in the peri-macular region is thus an attractive strategy to change the disease course of retinal dystrophies associated with FAM161A deficiency.

# Methods

## Reagents and tools table

| Reagent/resource | Reference or source | Identifier or catalog number |
|---|---|---|
| **Experimental models** | | |
| *Fam161A$^{tm1b/tm1b}$* (*M. musculus*) | Beryozkin et al, 2021 | |
| ARPE-19 | ATCC | CRL-2302 |
| hTERT-RPE1 | ATCC | CRL-4000 |
| 661 W | Tan et al, 2004 | |
| **Recombinant DNA** | | |
| *WPRE4* | Schambach et al, 2006 | |
| pEXPR gateway | Berger et al, 2015 | |
| pXR8 | Berger et al, 2015 | |
| **Antibodies** | | |
| *mAb=monoclonal; pAb=polyclonal* | | |
| Mouse mAb anti-acetylated-TUBULIN, ICC (1: 1000) | Sigma | T7451 |
| Mouse recombinant Ab anti α-TUBULIN U-ExM (1 :250) | ABCD antibodies, Lima and Cosson, 2019 | AA345-scFv-F2C |
| Mouse recombinant Ab β-TUBULIN U-ExM (1 :250) | ABCD antibodies, Lima and Cosson, 2019 | AA344-scFv-S11B |
| Rabbit pAb anti-CEP290 U-ExM (1: 250) | Proteintech | 22490-1-AP |
| Rabbit pAb FAM161A WB (for ARPE-19 blot 1:1000, for 661 W blot, 1:2000), ICC & IHC (1:500) | Sigma | HPA032119 |
| Rabbit pAb FAM161A, U-ExM (1:250) | ThermoFischer Scientific | PA5-56935 |
| Mouse mAb GADPH (clone 6C5) WB (for ARPE-19 blot 1:5000, for 661 W blot 1:1000) | Chemicon | MAB374 |
| Rabbit pAb GFP IHC (1:1000) | Abcam | Ab290 |

| Reagent/resource | Reference or source | Identifier or catalog number |
|---|---|---|
| Goat pAb anti-mouse Alexa Fluor 633 ICC, IHC, 1:2000 | ThermoFischer | A21053 |
| Goat pAb anti-mouse IgG HRP WB, 1:2000 | Amersham Biosciences | NA0931 |
| Goat pAb anti-rabbit Alexa Fluor 488 ICC, IHC, 1:2000 | ThermoFischer | A11070 |
| Goat pAb anti-Rabbit IgG HRP WB, 1: 2000 | Amersham Biosciences | NA0934 |
| Rabbit pAb anti-IFT81 U-ExM, 1:250 | Proteintech | 11744-1-AP |
| Rabbit pAb anti-LCA5 U-ExM, 1:250 | Proteintech | 19333-1-AP |
| Rabbit pAb anti-M/L-opsin U-ExM | Millipore | AB 5405 |
| Rabbit pAb anti-POC5 U-ExM, 1:250 | Bethyl | A303-341A |
| Mouse mAb anti-RHODOPSIN (clone Ret-P1) IHC, U-ExM | ThermoFischer Scientific | MA5-11741 |
| **Oligonucleotides and other sequence-based reagents** | | |
| PCR primers | This study | Appendix Table S1 |
| **Software** | | |
| GraphPad Prism (V10.0) | www.graphpad.com | |
| PickCentrioleDim plugin | Le Borgne et al, 2022 | |

## Mice

*Fam161A$^{tm1b/tm1b}$* mice (Beryozkin et al, 2021) were kept at 22 °C under a 12 h light/12 h dark cycle with the light on at 7 am and fed ad libitum.

## Plasmid designs

All regulatory regions were amplified by PCR from 293T genomic DNA (Appendix Table 1). Subcloning in pGEMT allowed to fuse IRBP (240 bp, position 1175–1415 Genbank X53044.1) to GRK1 (298 bp, 1760–2088 from AY32780.1), FCBR2 (456 bp, position 10,338–10,794 Genbank NG_028125.2) to F0.4 (459 bp, position 4554–5013 Genbank NG_028125.2) and to subclone FCBR1-F0.4 (1098 bp, position 3949–5013 of Genbank NG_028125.2). Human *FAM161A* short (HS, 1969 bp, position 20–1989 of Genbank sequence NM_032180.3) and long isoform (HL, 2150 bp, position 20–2170 of Genbank sequence NM_001201543.2) cDNAs were obtained after amplification from human retinal cDNA and cloning of the PCR product. A mutated Woodchuck hepatitis virus Posttranscriptional regulatory Element (WPRE4) devoid of promoter activities or open-reading frames (Matet et al, 2017; Schambach et al, 2006) was added in 3' of both cDNAs and 3XFlag tag (DYKDHDGDYKDHDIDYKDDDDK) was added in 5' of the cDNA when preparing the FCBR1-F0.4 and FCBR2-F0.4 plasmid vectors. The regulatory cassettes and the transgene cassettes were then assembled using the multiple insert gateway system to generate pEXPR bearing the ITR for AAV vector production (Berger et al, 2015).

For the in vitro study, the ubiquitous elongation short promoter was amplified from pLOX-GFP plasmid (Kostic et al, 2003) and

assembled with HL-WPRE4 or HS-WPRE4 cassettes using the multiple insert gateway system.

## Vector production

AAV viral vectors were produced by calcium chloride co-transfection of 293T cells of the transgene plasmid with the AAV packaging plasmid, a helper plasmid, and the pXR8 capsid plasmid as described in Berger et al (Berger et al, 2015). Three days after transfection, viral particles were harvested from supernatant and cell lysate, treated with DNase (#11284.932.001, Roche), and purified on an iodixanol density gradient (OptiPrep Density Gradient Medium, #D1556, SIGMA). Finally, desalting, purification, and concentration of the vectors were performed on Amicon Ultra-15 (#PL100 MILLIPORE) to resuspend the vectors in PBS + 0.001% Pluronic F-68 (# P5556, SIGMA). The concentration of the vector preparation was determined by quantitative PCR as described in Aurnhammer et al (Aurnhammer et al, 2012) and expressed in viral genome copy per ml (GC/ml).

## Vector injections

For surgery, postnatal day (PN) 14 to PN15 *Fam161A^tm1b/tm1b* mice were anesthetized by intraperitoneal injection of Ketamine (60 mg/kg) and Medetomidine (1 mg/kg). Pupils were dilated with single eye drops of tropicamide 1% and phenylephrine hydrochloride 2.5%. The eye was maintained exorbitated and perforated in the sclera at the limit of the limbus with a 30 G beveled Hamilton needle (nasal dorsal side). Then, under visualization using Viscotears (Ciba Vision) gel on which is placed a 6-mm-diameter coverslip, a Hamilton syringe with a blunt 34-G needle was introduced through this hole in the posterior chamber to reach the opposite eye cup site (temporal dorsal side) and create a subretinal detachment with 1 μl of viral solution. The needle was held for 1 min in this position before a careful extraction for the globe to minimize reflux. After surgery, the effects of anesthesia were reversed by injection of Atipazemole (1 mg/kg), and 25 μl Rimadyl (Zoetis) were injected subcutaneously. Rimadyl injections were repeated up to three times per 24 h the following day if needed.

## Electroretinogram recording

Mice were dark-adapted overnight, anesthetized with a mixture of Ketamine (133 mg/kg, Streuli, Uznach) and Xylazine (20 mg/kg, Bayler), and pupils dilated with a single eye drop of tropicamide 1% (Théa) and phenylephrine hydrochloride 2.5% (Bausch and Lomb). The animals were kept on a heating pad connected to a temperature control unit to maintain the temperature at 37 °C throughout the experiment. Responses to standard single light flashes (520 nm; half-bandwidth 35 nm) at 0.0001, 0.001, 0.01, 0.03, 0.1, 0.3, 1, 3, 10, and 30 cds/m$^2$ for scotopic ERG and after 10 min of light exposure to 3 cds/m$^2$ to 0.01, 0.03, 0.1, 0.3, 1, 3, 10 and 30 cds/m$^2$ for photopic ERG were generated by a stroboscope (Ganzfeld stimulator, Espion E3 apparatus; Diagnosys LLC). Retinal activity was recorded with a corneal electrode for each eye. The *a*-wave (photoreceptor-driven first negative wave) amplitude was measured from baseline to the bottom of the *a*-wave trough, and the *b*-wave (second order neuron driven, first positive wave) amplitude was measured from the bottom of the *a*-wave trough to the *b*-wave peak.

## Cell culture

For immunocytochemistry, ARPE-19 cells were seeded in DMEM/F12 (#11330-032, Gibco) supplemented with 10% FBS one day prior transfection while hTERT-RPE1 cells were seeded in DMEM/F12 supplemented with 1% FBS one day prior transfection on coverslips in a 24-well plate. Cells were transfected for overnight with 1 μg/well of pEXPR-EFS-HL-WPRE4 or pEXPR-EFS-HS-WPRE4 using Lipofectamine™ 3000 (ThermoFischer Scientific, reagent and protocol (day 1). Fresh media was applied at day 2 and day 4 to maintain transfected cells 5 more days in culture.

For western blot analysis, ARPE-19 cells and 661 W cells were seeded in a six-well plate in DMEM/F12 (#11330-032, Gibco) supplemented with 10% FBS, 1% penicillin/streptomycin and 0.5 mM β-mercaptoethanol, respectively. One day later, the cells were transfected with 2.5 μg/well of pEXPR-EFS-HL-WPRE4 or pEXPR-EFS-HS-WPRE4 using Lipofectamine™ 2000 reagent and protocol with the ratio 3:1 lipofectamine:DNA.

## Immunochemistry

### Labeling of cryosections
After enucleation, eyes were incubated at 4 °C in 4% paraformaldehyde for 30 min, washed twice with PBS, and incubated sequentially for 2 h in 10% and 20% sucrose, and finally overnight in 30% sucrose. Eyes were embedded in yazzulla (30% egg albumin (Applichem) and 3% gelatin in water) and cut with a cryostat in 14-μm sections. For all stainings, blocking was performed at RT for 1–1.5 h, the first antibody incubated at 4 °C overnight, and the secondary antibody for 1 h at RT (Reagents and Tools Table). The blocking solution is PBS supplemented with 1% bovine serum albumin (BSA) and 0.1% Triton X-100 for FAM161A antibody or with 10% normal goat serum (Dako) and 0.1% Triton X-100 for GFP antibody. Secondary antibodies Alexa Fluor 488 or 633 goat anti-rabbit or goat anti-mouse depending on the stainings were diluted to 1:2000 in PBS and counterstaining was finally performed with DAPI. Sections were mounted in MOWIOL.

### Immunocytochemistry
Coverslip were fixed for 10 min in 4% paraformaldehyde washed with PBS and blocked with 5% BSA and 0.1% Triton X-100 in PBS for 1 h at RT. Primary antibodies against FAM161A, GFP and acetylated tubulin (Reagents and Tools Table) were incubated in the blocking solution overnight at 4 °C. Secondary antibodies anti-rabbit-Alexa Fluor 488 and anti-mouse-Alexa Fluor 633 (Reagents and Tools Table) were incubated for 1 h at RT. DAPI was used as a counterstain.

## Ultrastructure expansion microscopy

### Expansion microscopy of frozen sections
Protocol for frozen section expansion was adapted from the U-ExM method (Gambarotto et al, 2019) as follows and is depicted in Appendix Fig. S6. First, frozen sections on slides were thawed at room temperature (RT) for 2 min. Then, a double-sided sticky spacer of 0.3-mm thickness (IS317, SunJin Lab Co.) was stuck around the section of interest. Then, the crosslinking prevention step was performed by incubation with 300 μL of 2% acrylamide (AA; A4058, Sigma-Aldrich) + 1.4% formaldehyde (FA; F8775,

Sigma-Aldrich) at 37 °C for 3 h. Then, after removing the first solution, the gelation step was done by adding 130 μL of monomer solution composed of 75 μl of sodium acrylate (stock solution at 38% (w/w) diluted with nuclease-free water, 408220, Sigma-Aldrich), 37.5 μl of AA, 7.5 μl of N, N′-methylenbisacrylamide (BIS, 2%, M1533, Sigma-Aldrich), 15 μl of 10× PBS together with ammonium persulfate (APS, 17874, Thermo Fisher Scientific) and tetramethylethylenediamine (TEMED, 17919, Thermo Fisher Scientific) as a final concentration of 0.5% for 1 h and 30 min at 37 °C. A 24-mm coverslip was added on top to close the chamber. Next, the coverslip was removed, and an open 15-mL falcon cap was stuck on top of the spacer to create an incubation chamber. 2 ml of denaturation buffer (200 mM SDS, 200 mM NaCl, 50 mM Tris Base in water (pH 9)) were added in the chamber and the slide was put for 30 min on top of a heat block warmed at 95 °C, allowing the gel to detach from the glass slide. Once detached, the gel was carefully transferred into a round bottom glass tube in 5 mL of denaturation buffer, and incubated for another 1 h and 30 min at 95 °C. Finally, the gel was rinsed and expanded in three successive ddH$_2$O baths, before processing to the immunostainings.

### Immunostainings of expanded frozen sections

After expansion, expanded gels were shrunk with three 5-min baths of 1× PBS. Then, primary antibodies were incubated overnight at 4 °C in PBS with 2% of bovine serum albumin (BSA). Gels were washed three times 5 min in PBS with 0.1% Tween 20 (PBST) prior to secondary antibodies incubation for 3 h at 37 °C. After a second round of washing (three times 5 min in PBST), gels were expanded with three 15-min baths of ddH20 before imaging. Antibodies used are referenced in the Reagents and Tools Table. Image acquisition was performed on an inverted Leica Thunder DMi8 microscope using a 20× (0.40 NA) or 63× (1.4 NA) oil objective with Thunder SVCC (small volume computational clearing) mode at max resolution, adaptive as "Strategy" and water as "Mounting medium" to generate deconvolved images. To avoid gel drifting during acquisition, gels were imaged on Poly-D-lysine (A3890401, Gibco)-coated 24-mm coverslips (0117640, Marienfeld).

### Antibody stripping

In order to have the localization of several proteins within the same precious sample, we used antibody stripping to allow several rounds of immunostainings. Briefly, stained and imaged gels were washed three times 5 min in PBST to allow their shrinkage. Then, gels were incubated in 1 mL of stripping buffer made of 200 μL SDS 10% (2% final), 62 μL of Tris HCl pH 6.8 1 M, 8 μL of ß-mercaptoethanol and 730 μL of pure water for 30 min at 50 °C. Then gels were washed three times 5 min in PBST allowing a new round of antibody staining. Image acquisition is done without using Poly-D-lysine coated coverslips to avoid losing tissue on the coverslip. To correct potential drifting issues during acquisition, drift correction has been performed manually (<7 pixels correction).

### Expansion factor calculation

The expansion factor was calculated in a semiautomated way by comparing the full width at half maximum (FWHM) of photo-receptor mother centriole proximal tubulin signal with the proximal tubulin signal of expanded human U2OS cell centrioles using the PickCentrioleDim plugin described elsewhere (Mercey et al, 2022). Briefly, 48 photoreceptor mother centrioles FWHM were measured and compared to a pre-assessed value of U2OS centriole width (25 centrioles: mean = 231.3 nm). The ratio between measured FWHM and known centriole width gave the expansion factor.

### Connecting cilium length measurement

Knowing the expansion factor, protein signal lengths were measured using a segmented line drawn by hand (ImageJ) to fit with photoreceptor curvature, and corrected with the expansion factor.

## Western blot

Three days after transfection, ARPE-19 and 661 W cells were harvested and proteins were extracted using the RIPA buffer. For the western blot, 25 μg proteins were loaded into each lane, blotted with FAM161A antibody, and revealed with anti-rabbit-HRP 1:2000 (Reagents and Tools Table). Anti-GAPDH (Reagents and Tools Table) was used as a control.

## Statistics

Statistical analyses were performed using GraphPad Prism (V10.0), and $P < 0.05$ was considered significant. Quantification of the different parameters on immunolabeled sections or recording and quantification of retinal function were done in the blind.

For the comparison of ONL thickness or CC length between groups (Figs. 2, 4, and 5), we used the one-way ANOVA Kruskal–Wallis test followed by Dunn's test multiple comparison between the groups or, for the CC length, to compare treated groups with WT. To determine the difference of ONL thickness at different locations across the retina (Fig. 6A) we applied a two-way ANOVA followed by a Tukey's test to determine significant differences at specific locations between the FAM161A and GFP-treated eyes. For the comparison of ONL thickness based on the CC density category (Fig. 6B), we performed an ordinary one-way ANOVA followed by a Tukey's test for multiple comparisons. Finally, we did a linear regression between the ONL thickness and CC density of the different groups (Fig. 6C).

ERG a- or b-wave amplitudes of treated eyes were compared to the fellow eyes treated with the control vector by two-way ANOVA. When significance was reached Sidak's multiple comparison test was performed to compare the two groups for each stimuli intensity.

## Study approval

The animals were handled in accordance with the statement of the "Animals in Research Committee" of the Association for Research in Vision and Ophthalmology, with the Swiss Federal Animal Protection Act, and protocols were approved by the animal research local committee (VD1367).

## Graphics

The synopsis image was created with BioRender.com.

## The paper explained

### Problem

Retinitis pigmentosa-28 (RP28) is a recessive retinal disorder leading to blindness associated with FAM161A gene defects. FAM161A is expressed in the connecting cilium and maintains the photoreceptor structure necessary for the sensory function of the cell. Loss of FAM161A leads to cilium disorganization, photoreceptor cell death, and impaired vision. Two isoforms of FAM161A exist in the human retina. This study assesses the efficacy of these isoforms in restoring connecting cilia structure and improving retinal function post-gene transfer.

### Results

AAV vectors inducing widespread FAM161A protein in the entire photoreceptor cells, rescue photoreceptor survival but not function. Co-administration of both isoforms with AAV vectors, enabling precise FAM161A expression in the connecting cilium, promotes both retina survival and functional improvement.

### Impact

This work provides significant preclinical insight into RP28 gene therapy, emphasizing the importance of fine-tuning therapeutic gene expression tailored to cell physiology and disease traits for restoring retinal function. Such precision is crucial for secure gene therapy involving structural proteins like FAM161A.

# Data availability

This study includes no data deposited in external repositories.

# Peer review information

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

## Acknowledgements

The authors would like to thank Sylvain V Crippa, Yaël Alves, Dana Wanner and Catherine Martin for excellent technical assistance. This work was supported by the Swiss National Science Foundation (Sinergia Grant CRSII3_141814 to CR, YA, and DS), by the Foundation Provisu (to YA, to CK and VH), by the Fondation "En souvenir du Dr. Georges Borel" (CK), by the Fondation Asiles des Aveugles (YA), by the Novartis foundation for biomedical research (to YA).

## Author contributions

**Yvan Arsenijevic**: Conceptualization; Data curation; Formal analysis; Supervision; Funding acquisition; Validation; Visualization; Writing—original draft; Writing—review and editing. **Ning Chang**: Conceptualization; Formal analysis; Validation; Investigation; Visualization; Methodology. **Olivier Mercey**: Formal analysis; Investigation; Methodology; Writing—review and editing. **Younes El Fersioui**: Investigation; Visualization. **Hanna Koskiniemi-Kuendig**: Investigation. **Caroline Joubert**: Investigation. **Alexis-Pierre Bemelmans**: Resources. **Carlo Rivolta**: Resources; Funding acquisition. **Eyal Banin**: Resources. **Dror Sharon**: Resources; Funding acquisition; Writing—review and editing. **Paul Guichard**: Methodology; Writing—review and editing. **Virginie Hamel**: Funding acquisition; Methodology; Writing—review and editing. **Corinne Kostic**: Conceptualization; Data curation; Formal analysis; Supervision; Funding acquisition; Validation; Visualization; Writing—original draft; Writing—review and editing.

## Disclosure and competing interests statement

YA, CK, NC, and DS are inventors of a patent pending "GENE THERAPY FOR FAM161A-ASSOCIATED RETINOPATHIES AND OTHER CILIOPATHIES" (PCT/EP2023/069304). Otherwise no other financial competing interests exist for this study.

