## [Peer Review File · EMBO Molecular Medicine]

Fine-tuning FAM161A gene augmentation therapy to restore retinal function

Yvan Arsenijevic, Ning Chang, Olivier Mercey, Younes El Fersioui, Hanna Koskiniemi-Kuendig, Caroline Joubert, Alexis Bemelmans, Carlo Rivolta, Eyal Banin, Dror Sharon, Paul Guichard, Virginie Hamel, and Corinne Kostic

Corresponding authors: Corinne Kostic (Corinne.Kostic@fa2.ch) , Yvan Arsenijevic (yvan.arsenijevic@fa2.ch)

Review Timeline:

Transferred from Review Commons:	10th Jan 24
Editorial Decision:	31st Jan 24
Revision Received:	28th Feb 24
Accepted:	28th Feb 24

Editor: Zeljko Durdevic

Transaction Report:

Review
COMMONS

This manuscript was transferred to EMBO Molecular Medicine following peer review at Review Commons.

Review #1

1. Evidence, reproducibility and clarity:

Evidence, reproducibility and clarity (Required)

The manuscript "Fine-tuning FAM161A gene augmentation therapy to restore retinal function" submitted by Arsenijevic et al., describes gene therapy for RP28 caused by mutation in FAM161A in human. The authors worked on Fam161a-deficient mice by testing different isoforms, dose, promoter and enhancers to control the expression level and localization of the protein to functionally rescue the mice to prevent blindness. The tight control of protein expression is required for mutation in genes coding structural proteins in the retina.

The authors have clearly showed the optimized combination of conditions to restore function of Fam161atm1b/tm1b mice and also area of improvement to make.

****Comments****

Table

1. It would be nice to have a table of isoform, dose, promoter, enhancer and other conditions tested and the brief summary of phenotype as Table.

Discussion

1. This experiment was done on knockout condition but in real patient different form of mutant protein will exist in retinal tissue. Authors indicated that co-expression of short and long form of FAM161A worked better to rescue function. How would authors cope with interfering endogenous mutant protein in real patients?

2. Related to the first question, the expression of these retinal structural proteins will be different in mice and human. How would authors optimize the vector for human patient gene therapy?

2. Significance:

Significance (Required)

This is an important and excellent work showing tight control of expression is required for future retinal gene therapy.

3. How much time do you estimate the authors will need to complete the suggested revisions:

Estimated time to Complete Revisions (Required)

(Decision Recommendation)

Less than 1 month

Yes

Review #2

1. Evidence, reproducibility and clarity:

Evidence, reproducibility and clarity (Required)

Arsenijevic et al. investigated the therapeutic function of the FCBR1-F0.4 promoter-driven expression of both the short and long isoforms of human FAM161A. The results showed that this method not only repaired the disorganized connecting cilium but also restored the appropriate expression and localization of other proteins in the connecting cilium, thus restoring retinal function. Additionally, the study systematically evaluated the AAV dose, different promoters, and FAM161A isoforms' effects on retinal survival and function. Overall, the study is novel and robust. Here are some suggestions that may help improve the manuscript:

Scotopic and photopic ERG were performed to study retinal function. However, mouse behavior tests such as optomotor response should be employed to confirm vision restoration.

The immunostaining in Figure 3 has some noise. Filtering the blocking solution before use could improve the quality of the staining.

In Figure 5f, the data of wildtype mice should be included for comparison.

The cited paper, such as 'Garafalo AV, Cideciyan AV, Heon E, Sheplock R, Pearson A, WeiYang Yu C, Sumaroka A, Aguirre GD, and Jacobson SG. Progress in treating inherited retinal diseases: Early subretinal gene therapy clinical trials and candidates for future initiatives.

Prog Retin Eye Res. 2020;77(100827),' should be an original research paper, not a review article.

****Referees cross-commenting****

Agree with the comments addressed by Reviewer #1 and #3

2. Significance:

Significance (Required)

Overall, the manuscript is clear and interesting. I suggest a major revision for the manuscript.

3. How much time do you estimate the authors will need to complete the suggested revisions:

Estimated time to Complete Revisions (Required)

(Decision Recommendation)

Less than 1 month

No

Review #3

1. Evidence, reproducibility and clarity:

Evidence, reproducibility and clarity (Required)

This manuscript led by Arsenijevic and Chang is an important technical development to the ocular gene therapy space, and touches on the important aspect of structural protein restoration by gene therapy, that is, the precise control of localization and subsequent functional realization. Overall the manuscript is well written, and the experiments are technically sound, with limitations acknowledged.

To briefly summarize, the authors wanted to understand precise control of FAM161A expression and connecting cilium (CC) restoration. They built on, and extended their previous work that showed limited structural and functional rescue by photoreceptor expression of the longer isoform of mouse FAM161A in Fam161a KO driven by IRBP-GRK1 promoter. In the current work, the authors experimented with delivering human ortholog of FAM161A cDNA, short, or long, or both isoforms using newly devised, relatively weak promoters. The main readouts include retinal morphology (e.g., ONL thickness), ERG, and protein localization by IHC (e.g., correct location, no ectopic expression). It is worth noting that the authors highlighted the use of expansion microscopy technology to examine the connecting cilium (CC) organization and protein expression, which may minimize the use of TEM for CC structure determination and enable acceleration.

My enthusiasm for recommending it for publication is high. Nonetheless, I have the following comments, hoping the authors could address to further improve the manuscript.

****Major:****

Fig 1A-B. Do hTERT-RPE1 cells endogenously express FAM161A? This set of images lacks a negative control (i.e., no transfected RPE1 cells). Western blot of FAM161A is recommended, similar to Fig 1C.

Fig 1C. The authors noted in the discussion that HS isoform is more abundant than HL isoform from human retinal extract. Although this is from 661W, a mouse photoreceptor cell line, it seems this is aligned with the notion. To echo with the last comment, I am curious to see if under the same transfection, the HS isoform is preferentially expressed in hTERT-RPE1 cells..

Fig 3 and Fig 5: low mag WT images of FAM161A are the same. But higher mag images (presumably selected from ROIs in low mag) are not the same. Please make sure of no duplication images.

Fig 4H. HS+HL combo, and HL alone, showed almost a polarized quantification, quite variable. Can the authors speculate the reason? Also can the authors comment on if there is any associated notable inflammation especially in high tier dosage (10^{11} GC)?

Can the authors comment on the difference in the injection time, PN14-15 in this study vs. PN24-29 in their previous study? Have the authors attempted to treat the older mice with the optimized vector?

Can the authors speculate on why IRBP-GRK1 human FAM161A did not realize functional rescue (Fig 2) as it did with mouse FAM161A (previous work)?

****Minor:****

While the manuscript is overall well communicated, there are areas requiring further proofread. For example, in the Abstract section, "In 15 years" should be "For 15 years", "14-days FAM161atm1b/tm1b mice" should be "14-day old". In the Introduction, "... suggesting that protein miss-localization" should be "mis-localization". In the last paragraph before Discussion, "(iii) the restauration of CC..." should be "restoration", etc.

I recommend the authors to use a table to summarize different promoters, titers and key findings (e.g., expression level, localization) used and refer back to each figure. Scale bars on all figures, or every set of images.

****Referees cross-commenting****

To reviewer #2, Fig5f - WT data was shown as the gray horizontal line. I had the same question but then saw they noted in the legends. I think it is fine to cite the PRER review article to make their point.

I agree with the comments addressed by Reviewer #1 and am glad we both raise the point of using table for summarization.

2. Significance:

Significance (Required)

This well-drafted paper represents a technical development that could supplement current gene therapy strategies to certain ciliopathies. In this particular case, the authors chose FAM161A, a disease causal gene to retinitis pigmentosa-28 and encodes for a microtubule-associated ciliary protein involved in organizing the connecting cilium in photoreceptors. Of importance, the authors devised novel promoters to drive gene expression and took advantage of expansion microscopy for quickly examining cilia proteins and structures. Conceptually, the techniques developed in this manuscript could be applicable to several other inherited retinal dystrophies that share similar disease mechanisms.

3. How much time do you estimate the authors will need to complete the suggested revisions:

Estimated time to Complete Revisions (Required)

(Decision Recommendation)

Less than 1 month

Yes

Full Revision

Manuscript number: RC-2023-02203

Corresponding author(s): Yvan, Arsenijevic and Corinne, Kostic

[Please use this template only if the submitted manuscript should be considered by the affiliate journal as a full revision in response to the points raised by the reviewers.]

*If you wish to submit a preliminary revision with a revision plan, please use our "Revision Plan" template. **It is important to use the appropriate template to clearly inform the editors of your intentions.**]*

1. General Statements [optional]

This section is optional. Insert here any general statements you wish to make about the goal of the study or about the reviews.

We would like to extend our warmest thanks to the reviewers for their constructive comments and strong support for our study.

This section is mandatory. Please insert a point-by-point reply describing the revisions that were already carried out and included in the transferred manuscript.

Reviewer #1:

Table

1. It would be nice to have a table of isoform, dose, promoter, enhancer and other conditions tested and the brief summary of phenotype as Table.

We thank the reviewer for this valuable suggestion and have now included a summary Table (Table 1) cited in the last result section.

Discussion

1. This experiment was done on knockout condition but in real patient different form of mutant protein will exist in retinal tissue. Authors indicated that co-expression of short and long form of FAM161A worked better to rescue function. How would authors cope with interfering endogenous mutant protein in real patients?

We thank the reviewer for raising this interesting point. Most mutations described so far are nonsense or frameshift mutations common to both long and short isoforms which, consequently, are not present at the protein level (Beryozkin et al 2020, doi.org/10.1038/s41598-020-72028-0, Matsevich et al 2022, doi.org/10.1016/j.xops.2022.100229). Thus, we don't expect to have an imbalance between the remaining functional alleles and the therapeutic ones. However, we cannot exclude the discovery of missense mutations and the effect of such allele would have to be molecularly evaluated to determine if

gene replacement is limited for this specific condition. This question could be assessed in cellular models by co-expression of both mutated and WT-tagged proteins or in organoid models.

2. Related to the first question, the expression of these retinal structural proteins will be different in mice and human. How would authors optimize the vector for human patient gene therapy?

Aware that the 60% homology between the human and mouse protein could cause important limitations for the evaluation of the vector in the mouse model, we are continuing the validation of our vectors in human retina organoids. We plan to test both the reliable localization of the human isoforms in WT organoid and the rescue of structural photoreceptor defects of FAM161A-deficient human organoids. In parallel, vector-derived expression will also be validated in non-human primates.

Reviewer #2:

Scotopic and photopic ERG were performed to study retinal function. However, mouse behavior tests such as optomotor response should be employed to confirm vision restoration.

In our hand, we didn't notice a significant modification of the optomotor response, and consequently of the estimated visual acuity, in *Fam161a^{tmb/tmb}* mice at 3.5 months corresponding to the endpoint of our study (see figure below). In a separate study to this work, we are thus conducting a follow-up long term gene therapy study to be able to complete the functional analysis of the gene therapy rescue with the optomotor response at age with significant decreased visual acuity in untreated mice compared to WT. We will have to wait at least 6 months to expect to see a difference between groups.

Figure Visual acuity changes with age in *Fam161a^{tmb/tmb}* mice (n=6-9).

The immunostaining in Figure 3 has some noise. Filtering the blocking solution before use could improve the quality of the staining.

Full Revision

We thank the reviewer for this suggestion. The blocking solution was already filtered and the limited success of the mouse *FAM161A* staining is due to the imperfect recognition of anti-human *FAM161A* antibodies to the mouse protein.

In Figure 5f, the data of wildtype mice should be included for comparison.

As noted by reviewer 3, in Fig5 F, the plain gray horizontal line surrounded by the 2 dotted ones are referring to the mean +/- SEM of the WT value respectively. We added "WT" on the right of the graph to highlight the plain line.

The cited paper, such as 'Garafalo AV, Cideciyan AV, Heon E, Sheplock R, Pearson A, WeiYang Yu C, Sumaroka A, Aguirre GD, and Jacobson SG. Progress in treating inherited retinal diseases: Early subretinal gene therapy clinical trials and candidates for future initiatives. Prog Retin Eye Res. 2020;77(100827),' should be an original research paper, not a review article.

As noted by reviewer 3, we think appropriate to cite this review which is a complete reference to the different gene therapy approaches developed for inherited retinal diseases.

Major:

Fig 1A-B. Do hTERT-RPE1 cells endogenously express FAM161A? This set of images lacks a negative control (i.e., no transfected RPE1 cells). Western blot of FAM161A is recommended, similar to Fig 1C.

We previously showed that hTERT-RPE1 cells express *FAM161A* in the basal body of the centriole (Di Gioia 2015), but we recognized that it is not apparent in Figure 1A and B, probably due to a limitation of the antibody reactivity which labeled only overexpressed proteins. We thus performed additional experiments using the human ARPE19 cell line to demonstrate endogenous *FAM161A* expression in untransfected cells and to perform a Western blot from human transfected cells. We observed that in untransfected cells *FAM161A* labeling is weak and is only revealed in the centriole labeled by centrin after a long exposure time (Figure 1A). When *FAM161A* HS or HL is overexpressed the *FAM161A* labeling is present in the cell body, very strong, and is observed with short exposure time (Figure 1A). We also extracted protein from untransfected and HS- or HL-transfected ARPE-19 cells to identify the *FAM161A* protein by Western blot (Figure 1B). Thus, we added the negative control and a western blot from human cells to answer reviewer comments.

Fig 1C. The authors noted in the discussion that HS isoform is more abundant than HL isoform from human retinal extract. Although this is from 661W, a mouse photoreceptor cell line, it seems this is aligned with the notion. To echo with the last comment, I am curious to see if under the same transfection, the HS isoform is preferentially expressed in hTERT-RPE1 cells.

We do not think that transfection experiment is sufficient to prove that HS is preferentially expressed than HL. Even if we transfect the same amount of DNA, we would need an internal control for transfection to allow relative quantification of the protein expression after transfection. However, we performed an additional experiment in human RPE cells using the ARPE-19 cell line which is more efficiently transfected than hTERT-RPE1 in our hands. As shown in Figure 1B, we observed again more abundant expression of HS in these human transfected cells. However, we cannot exclude difference in

transfection efficiency between HL and HS conditions that could explain the difference in the final amount of FAM161A protein.

Fig 3 and Fig 5: low mag WT images of FAM161A are the same. But higher mag images (presumably selected from ROIs in low mag) are not the same. Please make sure of no duplication images.

We are facing technical limits with the labeling of the mouse Fam161A. The antibodies available have limited affinity for the mouse Fam161A protein. While we were able to perform Uex-M from mouse tissue samples (flatmount retina) to study Fam161A expression in the connecting cilium (Mercey et al PLoS Biol 2022), it was more challenging to obtain low magnification picture from mouse retina sections. We propose to show in Figure 3 mouse Fam161A expression obtained from retina section and keep the low magnification from a flatmount for the figure 5. Thus, there will be no duplication of images as recommended by the reviewer.

Fig 4H. HS+HL combo, and HL alone, showed almost a polarized quantification, quite variable. Can the authors speculate the reason?

Despite the fact that injections are targeting similar retinal region in treated animals, there is still variation in the localization and extend of the gene transfer due to the surgical success. Indeed, the area of retinal detachment is hard to control in the mouse as of the quality of re-attachment. Moreover, the effective dose may lightly vary when some viral particles might be loss due to reflux. One would need to treat a larger number of eyes to really conclude that HS alone would be less variable than HL alone or HS+HL. However, we could also speculate that HS+HL and HL treatments being more efficient to rescue connecting cilium length compared to HS alone (Fig 5F) could, in the best injected eyes, have a better ONL thickness rescue than the limited ONL rescue induced by HS treatment.

Also can the authors comment on if there is any associated notable inflammation especially in high tier dosage (10^{11} GC)?

We didn't follow inflammation directly by fundus examination or OCT imaging following injection. However, despite the high dose used in our successful conditions ($10E11$ GC/eye), we didn't notice any differences in the general mouse welfare after injection compare to lower doses. Systemic administration of Rimadyl (carprofen) was however adapted to each mouse during the 24 hrs post-surgery. In comparison to other groups with lower vector doses, no particular emergence of inflammatory cells or damages were observed by histology.

Can the authors comment on the difference in the injection time, PN14-15 in this study vs. PN24-29 in their previous study? Have the authors attempted to treat the older mice with the optimized vector?

The gene therapy study using the mouse cDNA was performed before establishing the time course of connecting cilia disruption in the *Fam161a*^{tm^b/tm^b} mouse (Mercey et al. 2022). Following the observation that CC develop similarly to healthy animal up to postnatal day 10, we decided to treat the mouse earlier for the second gene therapy study using human proteins. Nonetheless, the action of the vector occurred when the cilium is already disorganized as we expect expression of the WT Fam161A from 2 weeks post-injection. We are now testing treatments at different ages, including PN28, to determine the therapeutic window and if the optimal conditions (dose, ratio) may vary with the age at treatment.

Can the authors speculate on why IRBP-GRK1 human FAM161A did not realize functional rescue (Fig 2) as it did with mouse FAM161A (previous work)?

Our hypothesis to explain the absence of functional rescue following IRBP-GRK1 vector injection is that the difference in human protein distribution compared to the mouse protein in the mouse retina could impact the function of the photoreceptor by interfering with physiological process such as transport. As mentioned in our discussion: “overexpression of these proteins could saturate the transport system impacting the cellular processes”.

As mentioned in the discussion, there is only 60% of homology between human and mouse proteins which could induce a major impact on protein localization and function. Post-translational modification which are also known to be crucial for modulating connecting cilium addressing (Rao et al. 2016) could also differ and impact both human protein distribution and function (for example 3 cysteines in the human protein sequence could be palmytoylated (C359, C366, C367) and are absent in the mouse sequence). Moreover, the exact role of the human long and short isoforms are unknown and their adaptability to the mouse system not yet identified. Further studies should be performed to understand the consequence of such differences on the function and to unravel the function of both long and short human isoforms in the retina.

Minor:

While the manuscript is overall well communicated, there are areas requiring further proofread. For example, in the Abstract section, "In 15 years" should be "For 15 years", "14-days FAM161atm1b/tm1b mice" should be "14-day old". In the Introduction, "... suggesting that protein miss-localization" should be "mis-localization". In the last paragraph before Discussion, "(iii) the restauration of CC..." should be "restoration", etc.

We corrected these errors and carefully proofread the whole manuscript to avoid typing mistakes.

I recommend the authors to use a table to summarize different promoters, titers and key findings (e.g., expression level, localization) used and refer back to each figure.

We thank the reviewer for this valuable suggestion and have now included a summary Table (Table 1) cited in the last result section.

Scale bars on all figures, or every set of images.

We added scale bars on figures containing microscopic images.

31st Jan 2024

Dear Dr. Kostic,

Thank you for the submission of your revised manuscript to EMBO Molecular Medicine. I am pleased to inform you that we will be able to accept your manuscript pending the following final amendments:

1) Please address referee #1 minor concern.

2) Figures: We note that some images/panels are reused. Figure 2I is reused in Appendix Figure S2. Please cite in the respective figure legend every reused image/panel.

3) Abstract: I have gone through your text and revised it (see below). Please review it, amend as you see fit:

For the past 15 years, gene therapy has been viewed as a beacon of hope for inherited retinal diseases. Many preclinical investigations have centered around vectors with maximal gene expression capabilities, yet despite efficient gene transfer, minimal physiological improvements have been observed in various ciliopathies. Retinitis pigmentosa-type 28 (RP28) is the consequence of bi-allelic null mutations in the FAM161A, an essential protein for the structure of the photoreceptor connecting cilium (CC). In its absence, cilia become disorganized, leading to outer segment collapses and vision impairment. Within the human retina, FAM161A has two isoforms: the long one with exon 4 and the short one without it. To restore CC in Fam161a-deficient mice, we compared AAV vectors with varying promoter activities, doses, and human isoforms. While all vectors improved cell survival, only the combination of both isoforms using the weak FCBR1-F0.4 promoter enabled precise FAM161A expression in the CC and enhanced retinal function. Our investigation into FAM161A gene replacement for RP28 emphasizes the importance of precise therapeutic gene regulation, appropriate vector dosing, and the delivery of both isoforms.

4) Author checklist: Please submit a complete checklist. <https://www.embopress.org/pb-assets/embosite/EMBO%20Press%20Author%20Checklist-1642513524327.xlsx>

5) In the main manuscript file, please do the following:

- Please address all comments suggested by our data editors listed below:

o Figure legends:

1. Please note that a separate 'Data Information' section is required in the legends of figures 1a, c, e; 2e-m; 4a-h; 5a-f, supplementary figures 3a-c.

2. Please note that the legends for figures 4c-d is not provided in the sequential manner (legend for figure 4d is provided before legend of figure 4c). This needs to be rectified.

3. Please define the annotated p values ****/***/**/* in the legend of figure 6a-b; 7a-c; as appropriate.

4. Please indicate the statistical test used for data analysis in the legends of figures 2a-d, n; 4h; 5f; 6a-b; 7e-g.

5. Please note that information related to n is missing in the legends of figures 2n; 4h; 5f; 6a-c; 7e-g.

6. Although 'n' is provided, please describe the nature of entity for 'n' in the legends of figures 7a-c.

7. Please note that the error bars are not defined in the legends of figures 2a-d; 4h; 6c; 7a-c; 7e-g.

8. Please note that the scale bar needs to be defined for figures 4a-g.

9. Please note that scale bar and its definition are missing for supplementary figures 2; 3a-c; 4; 5.

10. Please note that the white arrows are not defined in the legend of figure 2e. This needs to be rectified.

- Add up to 5 keywords.

- Add callouts for Fig 2E-F, Fig 4H, Fig 5E, Fig 7A-B.

- Remove "data not shown" (p. 8 and 10)

- Rename "Methods" to "Materials and Methods".

- Provide the antibody dilutions that were used for each antibody.

- Rename "Author disclosure statement" "Disclosure and competing interests statement". We updated our journal's competing interests policy in January 2022 and request authors to consider both actual and perceived competing interests. Please review the policy <https://www.embopress.org/competing-interests> and update your competing interests if necessary.

- Author contributions: Please remove it from the manuscript and specify author contributions in our submission system. CRediT has replaced the traditional author contributions section because it offers a systematic machine-readable author contributions format that allows for more effective research assessment. You are encouraged to use the free text boxes beneath each contributing author's name to add specific details on the author's contribution. More information is available in our guide to authors:

<https://www.embopress.org/page/journal/17574684/authorguide#authorshipguidelines>

- Correct the reference citation in the text and reference list. In the text a reference should be cited by author and year of publication. Include a space between a word and the opening parenthesis of the reference that follows. In the reference list, citations should be listed in alphabetical order. Where there are more than 10 authors on a paper, 10 will be listed, followed by "et al.". Please check "Author Guidelines" for more information.

- <https://www.embopress.org/page/journal/17574684/authorguide#referencesformat>

- Please add data availability statement. If no data are deposited in public repositories, add the sentence: This study includes no data deposited in external repositories.

Please check "Author Guidelines" for more information.

<https://www.embopress.org/page/journal/17574684/authorguide#availabilityofpublishedmaterial>

6) Appendix: Suppl. Figures and Tables should be placed in an "Appendix" - combined in a single PDF file, with a ToC with page numbers, legend underneath the figure, and the figures renamed to "Appendix Figure S1" etc and tables to "Appendix Table S1" etc. Please also update all figure and table callouts in the main text. For more information on figure presentation please check "Author Guidelines".

<https://www.embopress.org/page/journal/17574684/authorguide#datapresentationformat>

7) The Paper Explained: Please provide "The Paper Explained" and add it to the main manuscript text. Please check "Author Guidelines" for more information. <https://www.embopress.org/page/journal/17574684/authorguide#researcharticleguide>

8) Synopsis: Every published paper now includes a 'Synopsis' to further enhance discoverability. Synopses are displayed on the journal webpage and are freely accessible to all readers. They include separate synopsis image and synopsis text.

- Synopsis image: Please provide a striking image or visual abstract as a high-resolution jpeg file 550 px-wide x (250-400)-px high to illustrate your article.

- Synopsis text: Please provide a short standfirst (maximum of 300 characters, including space) as well as 2-5 one sentence bullet points that summarise the paper as a .doc file. Please write the bullet points to summarise the key NEW findings. They should be designed to be complementary to the abstract - i.e. not repeat the same text. We encourage inclusion of key acronyms and quantitative information (maximum of 30 words / bullet point). Please use the passive voice.

9) For more information: This space should be used to list relevant web links for further consultation by our readers. Could you identify some relevant ones and provide such information as well? Some examples are patient associations, relevant databases, OMIM/proteins/genes links, author's websites, etc...

10) As part of the EMBO Publications transparent editorial process initiative (see our Editorial at <http://embomolmed.embopress.org/content/2/9/329>), EMBO Molecular Medicine will publish online a Review Process File (RPF) to accompany accepted manuscripts. This file will be published in conjunction with your paper and will include the anonymous referee reports, your point-by-point response and all pertinent correspondence relating to the manuscript. Let us know whether you agree with the publication of the RPF and as here, if you want to remove or not any figures from it prior to publication. Please note that the Authors checklist will be published at the end of the RPF.

11) Please provide a point-by-point letter INCLUDING my comments as well as the reviewer's reports and your detailed responses (as Word file).

I look forward to reading a new revised version of your manuscript as soon as possible.

Yours sincerely,

Zeljko Durdevic

*** Instructions to submit your revised manuscript ***

- 1) a .docx formatted version of the manuscript text (including Figure legends and tables)
- 2) Separate figure files*

3) supplemental information as Expanded View and/or Appendix. Please carefully check the authors guidelines for formatting Expanded view and Appendix figures and tables at <https://www.embopress.org/page/journal/17574684/authorguide#expandedview>

4) a letter INCLUDING the reviewer's reports and your detailed responses to their comments (as Word file).

5) The paper explained: EMBO Molecular Medicine articles are accompanied by a summary of the articles to emphasize the major findings in the paper and their medical implications for the non-specialist reader. Please provide a draft summary of your article highlighting

6) For more information: There is space at the end of each article to list relevant web links for further consultation by our readers. Could you identify some relevant ones and provide such information as well? Some examples are patient associations, relevant databases, OMIM/proteins/genes links, author's websites, etc...

7) Author contributions: the contribution of every author must be detailed in a separate section.

8) EMBO Molecular Medicine now requires a complete author checklist (<https://www.embopress.org/page/journal/17574684/authorguide>) to be submitted with all revised manuscripts. Please use the checklist as guideline for the sort of information we need WITHIN the manuscript. The checklist should only be filled with page numbers where the information can be found. This is particularly important for animal reporting, antibody dilutions (missing) and exact values and n that should be indicated instead of a range.

9) Every published paper now includes a 'Synopsis' to further enhance discoverability. Synopses are displayed on the journal webpage and are freely accessible to all readers. They include a short stand first (maximum of 300 characters, including space) as well as 2-5 one sentence bullet points that summarise the paper. Please write the bullet points to summarise the key NEW findings. They should be designed to be complementary to the abstract - i.e. not repeat the same text. We encourage inclusion of key acronyms and quantitative information (maximum of 30 words / bullet point). Please use the passive voice. Please attach these in a separate file or send them by email, we will incorporate them accordingly.

You are also welcome to suggest a striking image or visual abstract to illustrate your article. If you do please provide a jpeg file 550 px-wide x 300-800px high.

10) A Conflict of Interest statement should be provided in the main text

11) Please note that we now mandate that all corresponding authors list an ORCID digital identifier. This takes <90 seconds to complete. We encourage all authors to supply an ORCID identifier, which will be linked to their name for unambiguous name identification.

Currently, our records indicate that the ORCID for your account is 0000-0003-1006-9733.

Link Not Available

- Graphs 800-1,200 DPI
- Photos 400-800 DPI
- Colour (only CMYK) 300-400 DPI"

*Additional important information regarding figures and illustrations can be found at <https://bit.ly/EMBOPressFigurePreparationGuideline>. See also figure legend preparation guidelines:

**** Reviewer's comments ****

Referee #1 (Comments on Novelty/Model System for Author):

This study is carefully designed and well executed, and the manuscript is clearly communicated. I think it would be a good addition to the ocular gene therapy space and may serve precedents to other IRDs that may share similar mechanisms.

Referee #1 (Remarks for Author):

All my comments have been addressed. Minor comment: the molecular ladder for APRE19 and 661W cells seem to differ, thus suggesting the HS vs. HL isoforms showed different molecular weight. Can the authors confirm?

Referee #2 (Remarks for Author):

I have no further recommendation.

Rev_Com_number: RC-2023-02203

New_manu_number: EMM-2024-19281

Corr_author: Kostic

Title: Fine-tuning FAM161A gene augmentation therapy to restore retinal function

The authors addressed the remaining editorial issues.

28th Feb 2024

Dear Dr. Kostic,

We are pleased to inform you that your manuscript is accepted for publication and is now being sent to our publisher to be included in the next available issue of EMBO Molecular Medicine.
